



# Measurement report: High Contributions of Halohydrocarbon and Aromatic Compounds to Emissions and Chemistry of Atmospheric VOCs in Industrial Area

Ahsan Mozaffar [1,2,3], Yan-Lin Zhang [1,2,3*], Yu-Chi Lin[1,2,3],  Feng Xie [1,2,3], Mei-Yi Fan [1,2,3], and Fang Cao [1,2,3]

5 [1]Yale-NUIST Center on Atmospheric Environment, International Joint Laboratory on Climate and Environment Change, Nanjing University of Information Science and Technology, Nanjing, 210044, China.

[2]Key Laboratory Meteorological Disaster; Ministry of Education & Collaborative Innovation Center on Forecast and Evaluation of Meteorological Disaster, Nanjing University of Information Science and 10 Technology, Nanjing, 210044, China.

[3]Jiangsu Provincial Key Laboratory of Agricultural Meteorology, College of Applied Meteorology, Nanjing University of Information Science & Technology, Nanjing 210044, China.

*Correspondence to*: Yan-Lin Zhang (dryanlinzhang@outlook.com)

**Abstract.** Volatile organic compounds (VOCs) are key components for tropospheric chemistry and air 15 quality. We investigated ambient VOCs in an industrial area in Nanjing, China from July 2018 to May 2020. The total VOCs (TVOCs) concentration was 59.8±28.6 ppbv during the investigation period. About twice TVOCs concentrations were observed in autumn (83±20 ppbv) and winter (77.5±16.8 ppbv) seasons compared to those in spring (39.6±13.1 ppbv) and summer (38.8±10.2 ppbv). Unlike previous studies in Nanjing, oxygenated-VOCs (OVOCs) and halohydrocarbons were measured, the 20 observed TVOCs was about 1.5 and 3-times higher than those previously reported in the same study area and a nonindustrial suburban area in Nanjing, respectively. Observed TVOCs concentrations were similar to those in metropolitan city Beijing and Shanghai, however, it was about 1.5-3 folds higher than those in Lanzhou, Wuhan, Tianjin, Ningbo, Chengdu, London, Los Angeles, and Tokyo. Due to the industrial influence, halohydrocarbons (14.3±7.3 ppbv, 24%) VOC-group was the second largest 25 contributor to the TVOCs after alkanes (21±7 ppbv, 35%), which is in contrast with the previous studies in Nanjing and also in almost other regions in China. Relatively high proportions of helohydrocarbons



and aromatics were observed in autumn (25.7 and 19.3%, respectively) and winter (25.8 and 17.6%, respectively) compared to those in summer (20.4 and 11.8%, respectively) and spring (20.3 and 13.6%, respectively). According to the potential source contribution function (PSCF), short-distance transports from the surrounding industrial areas and cities were the main reason for high VOC concentration in the study area. According to positive matrix factorization (PMF) model results, industry-related sources (23-47%) followed by vehicle emissions (24-34%) contributed the major portion to the ambient VOC concentrations. Whereas aromatics followed by alkenes were the top contributors to the loss rate of OH radicals ($L^{OH}$) (37 and 32%, respectively), alkenes followed by aromatics contributed most to the ozone formation potential (OFP) (39 and 28%, respectively). Besides, the aromatics VOC-group was also the major contributor to the secondary organic aerosol potential (SOAP) (97%). According to the empirical kinetic modelling approach (EKMA) and relative incremental reactivity (RIR) analysis in assistance with a photochemical box model, the study area was in the VOC-sensitive regime for ozone ($O_3$) formation during all the measurements seasons. Therefore, mainly alkenes and aromatics emissions chiefly from industries and automobiles should be reduced to decrease the secondary air pollution formation in the study area.

## 1 Introduction

Air pollution characterized by severe ozone ($O_3$) and haze pollution is a big problem in urban and industrial areas in China (He et al., 2019; Hui et al., 2018; Tan et al., 2018;  Jia et al., 2016; Feng et al., 2016; Hui et al., 2019). In recent years, $O_3$ concentration above the national standard, and severe haze events are frequently reported (He et al., 2019; Hui et al., 2019; Sheng et al., 2018; Feng et al., 2016; Tan et al., 2018; Jia et al., 2016). As a precursor of $O_3$ and secondary organic aerosol (SOA), volatile organic compounds (VOCs) are largely responsible for the severe air pollution in China (Song et al., 2018; Hui et al., 2019; Hui et al., 2018; He et al., 2019). Unfortunately, anthropogenic VOC emissions have been increasing over the last 2 decades in China and it is expected to do so in the future (Mozaffar & Zhang, 2020, and references therein).

Atmospheric VOC has plenty of sources; it can be emitted from various anthropogenic and biogenic sources. Besides, it can also be formed in the atmosphere. Anthropogenic VOC sources mainly include



industrial emission, vehicle exhaust, solvent usages, biomass burning, and fuel evaporation. On the
other hand, vegetation is the main biogenic sources of VOC. In developed areas in China, vehicle
exhaust and industrial emission are the 2 major VOC sources (He et al., 2019; Hui et al., 2018; Hui et
al., 2019; Mo et al., 2017; Song et al., 2018; An et al., 2014; Mozaffar & Zhang, 2020). Whereas
vehicle-related sources are more dominant in the North China Plain (NCP), Central China (CC), and
Pearl River Delta region (PRD), industry-related sources are more influential in the Yangtze River Delta
(YRD) area (Zhang et al., 2017; Meng et al., 2015;  Sun et al., 2019; He et al., 2019; Zhang et al., 2018;
An et al., 2017; Mozaffar & Zhang, 2020; Shao et al., 2016). Alkanes, Alkenes, aromatics, oxygenated-
VOCs (OVOCs), and halohydrocarbons are the most common VOC-groups in the atmosphere (Hui et
al., 2019; Hung-Lung et al., 2007; Song et al., 2018; Tiwari et al., 2010; He et al., 2019; Na et al., 2001;
Hui et al., 2018). VOC concentration and composition changes depending on seasons, for example, the
contribution from biogenic and solvent utilization increases in summer, and contribution from
combustion sources increases in winter (Mo et al., 2017; Song et al., 2018; An et al., 2014). The
chemical reactivity of VOC depends on its chemical composition, for instance, alkenes and aromatics
are generally more reactive than alkanes (Carter, 2010). To understand the chemical reactivity and
secondary product formation ability of VOCs, analysis of OH radical loss rate ($L^{OH}$), ozone formation
potential (OFP), and secondary organic aerosol potential (SOAP) are commonly used (Song et al., 2018;
He et al., 2019; Hui et al., 2018; Hui et al., 2019).

Industries are an important source of VOC, and different reactive and hazardous VOCs emissions from
industries are already reported in different areas on earth (Zhang et al., 2018; Na et al., 2001; Hung-
Lung et al., 2007; Yan et al. 2016; Tiwari et al., 2010; Shi et al., 2015; Zhang et al., 2018b). For
instance, Zhang et al. (2018) reported a high concentration of alkanes (82%) and lifetime cancer risk of
different aromatics and halohydrocarbons in a petroleum refinery in Guangzhou, China. A high
concentration of OVOCs (63%) was observed in an industrial area in Ulsan, Korea (Na et al., 2001).
Hung-Lung et al.(2007) mentioned a high concentration of aromatics in an industrial area in Taiwan. A
high concentration of halohydrocarbons (49%) was observed in an iron smelt plant in Liaoning, China
(Shi et al., 2015).  Zhang et al. (2018) mentioned a high concentration of alkanes (42%) and aromatics
(20%) in a petrochemical and other industries affected area in Shanghai, China. High concentrations of





aliphatic and aromatics were observed in a petrochemical industrial area in Yokohama, Japan (Tiwari et al., 2010). Therefore, VOC composition varied among the industries/industrial areas in different regions. Mostly short-term investigations were performed to characterize the VOCs in industry-affected areas. In the current study, we carried out a comprehensive investigation on VOC in an industrial area in Nanjing between July 2018 and May 2020. Nanjing is located in the YRD region which is mainly affected by industrial emissions. Several VOC investigations have already been performed in the Nanjing industrial area but OVOCs and halohydrocarbons were not measured in those studies (An et al., 2017; An et al., 2014). However, OVOCs and halohydrocarbons are already mentioned as one of the highest concentrated VOC-groups in other industrial regions (Na et al., 2001; Shi et al., 2015). In the current study area, a high concentration of alkanes (45%) and alkenes (25%) were observed in a previous investigation (An et al., 2014). Besides the incomplete VOC measurements, $O_3$ formation sensitivity to its precursors was not investigated properly using a photochemical box model in Nanjing. Moreover, source apportionment of VOCs was not conducted for different seasons of a year.

In the current study, we report the variations in concentrations and compositions of VOC during the observation period. We present the possible source areas and potential sources of VOC based on potential source contribution function (PSCF) and positive matrix factorization (PMF) model analysis. We also present the contributions of different sources to ambient VOC during the measurement period. We report the chemical reactivity and secondary product formation capacity of the VOC using $L^{OH}$, OFP, and SOAP analysis. We also present the sensitivity analysis of $O_3$ formation using empirical kinetic modelling approach (EKMA) and relative incremental reactivity (RIR) analysis. Therefore, this study provides valuable information to the scientific community and policymakers.

## 2 Material and Methods

### 2.1 Sampling Site Description, Gases Analysis, and Meteorology Data

Field measurements were carried out from July 2018 to May 2020 at Nanjing University of Information Science and Technology (32.1°N, 118.4°E), which is located in an industrial area in Nanjing, China.



The sampling site was on the rooftop of a building (~20 m). The sampling site is surrounded by different chemical and petrochemical industries, steel plants, gas stations, high traffic roads, and residential areas. A detailed description of the sampling site can be found elsewhere (Mozaffar et al., 2020).

We analysed ambient air VOCs using an online GC-FID/MS instrument (AC-GCMS 1000, Guangzhou Hexin Instrument Co., Ltd., China). FID detector analysed C2-C5 VOCs and MS analysed C6-C12 VOCs. The instrument analysed one sample at every hour. During the investigation period, we inspected and calibrated the instrument regularly to ensure the accuracy of the data. We monitored the $O_3$ concentrations using a 49i $O_3$ analyser (Thermo Fisher Scientific Inc., USA), NO, $NO_2$ and NOx concentrations were measured using a 42i NO-$NO_2$-NOx analyser (Thermo Fisher Scientific Inc., USA), $SO_2$ concentrations were followed using a 43i $SO_2$ analyser (Thermo Fisher Scientific Inc., USA), and CO concentrations were measured using a 48i CO analyser (Thermo Fisher Scientific Inc., USA). We also measured temperature and relative humidity, wind speed, wind direction, and solar radiation by HMP155 (Vaisala, Finland), 010C (Met One Instruments, Inc., USA), 020CC (Met One Instruments, Inc., USA), and CNR4 (Kipp & Zonen, The Netherlands) analysers, respectively. A detailed description of the instrumentation, sampling procedure, analysis, quality control, and calibration procedure can be found elsewhere (Mozaffar et al., 2020).

## 2.2 Positive Matrix Factorization (PMF) model and Potential Source Contribution Function (PSCF)

We used the positive matrix factorization (PMF) model (US Environmental Protection Agency, USEPA, version 5.0) for the source apportionments of VOCs. A detailed description of the model can be found elsewhere (Hui et al., 2019; Song, Tan, Feng, Qu, Liu, et al., 2018). In this study, we used 62 potential VOC tracers (Fig. S1 - S4) in the PMF model to analyse the VOC sources for different seasons. The error fraction was set to 20% for the sample data uncertainty estimation. We explored the PMF factor number from 4-8 to determine the optimal number of sources. Finally, we decided to choose an 8-factor solution ($Q_{true}/Q_{robust}$ = ~1.0).

We used the potential source contribution function (PSCF) to locate possible source areas of VOCs for different seasons during the investigation period. We used Zefir analysis software to do the PSCF



analysis and the Hysplit4 model to cluster the backward trajectories (Petit et al., 2017). Backward trajectories in the sampling site were estimated using the data provided by the National Centers for Environmental Prediction (ftp://arlftp.arlhq.noaa.gov/pub/archives/gdas1). We estimated 24 hr backward trajectories 24 times a day arriving at 500 m above the ground surface using the hysplit4 model. For the PSCF analysis, we divided the geographic region covered by the back trajectories into an array of 0. 1° × 0. 1° grid cells and used the mean TVOCs concentration as the VOC reference value. More details about the PSCF analysis can be found in previous studies (Chen et al., 2018).

## 2.3 OH radical loss rate (L$^{OH}$), Ozone formation potential (OFP), and Secondary organic aerosol potential (SOAP)

To evaluate the daytime photochemistry of VOCs, we estimated their OH radical loss rate (L$^{OH}$). The following equation was used to estimate the L$^{OH}$ (s$^{-1}$) (Zhang et al., 2020).

$$L^{OH} = [VOC]_i \times K_i^{OH} \tag{1}$$

Where $[VOC]_i$ is the concentration of VOC species i (molecule cm$^{-3}$), $K_i^{OH}$ (cm$^3$ molecule$^{-1}$ s$^{-1}$) is the reaction rate constant of i VOC with OH radical. The K$^{OH}$ values for the VOCs are collected from Carter (2010) (Table S1).

The Ozone formation potential (OFP) of the VOCs is their maximum contribution to the O$_3$ formation (Hui et al., 2018a). The OFP (ppbv) of the VOCs was estimated using the following equation.

$$OFP = [VOC]_i \times MIR_i \tag{2}$$

Where MIR$_i$ is the maximum incremental reactivity of the i VOC. The MIR values for the VOCs are also collected from Carter (2010) (Table S1).

The contribution of VOCs to the formation of secondary organic aerosol is estimated by secondary organic aerosol potential (SOAP) (Song et al., 2018). We estimated the SOAP (ppbv) of VOCs using the following equation.

$$SOAP = [VOC]_i \times SOAP_i^p \tag{3}$$





Where $SOAP_i^p$ is the SOA formation potential of the i VOC on a mass basis relative to toluene (Derwent et al., 2010). In this study, the $SOAP^p$ factors of the VOCs are collected from Derwent et
al.(2010) (Table S1).

**2.4 Empirical Kinetic Modelling Approach (EKMA) and Relative Incremental Reactivity (RIR)**

The empirical kinetic modelling approach (EKMA) is a well-known procedure to develop the $O_3$ formation reduction strategy by testing the relationship between ambient $O_3$ and its precursors (He et al., 2019; Hui et al., 2018; Vermeuel et al., 2019; Tan et al., 2018). In this study, we used the
Framework for 0-D Atmospheric Model (F0AM v 3.2, Wolfe et al., 2016), a photochemical box model run by Master Chemical Mechanism (MCM) v3.2 chemistry (Jenkin et al., 1997; 2003, 2015; Saunders et al., 2003), to get the data for the EKMA isopleth. The FOAM-MCM box model can simulate 16940 reactions of 5733 chemical species. The box model was run using the VOCs and gas concentrations and the meteorological data as input. To generate the $O_3$ isopleth from the model simulated data, a total of
121 reduction scenarios (11 NOx ×11 VOC) were simulated and the maximum $O_3$ produced in each scenario was saved.

The $O_3$ formation sensitivity to its precursors' concentrations can also be assessed by the relative incremental reactivity (RIR, Cardelino & Chameides, 1995). We also utilized the FOAM-MCM box model data to estimate the RIR. The RIR is simply defined as the percentage change in $O_3$ formation per
percentage change in precursor's concentration. In this study, we reduced the precursor's concentration by 10% for the RIR estimation. The RIR was estimated using the following equation.

$$RIR\ (X) = \frac{[P_{O_3}(X) - P_{O_3}(X - \Delta X)]/P_{O_3}(X)}{[\Delta X]/[X]} \qquad (4)$$

Where [X] is the observed concentration of a precursor X, [ΔX] is the changes in the concentration of X. $P_{O3}(X)$ and $P_{O3}(X- \Delta X)$ are the simulated net $O_3$ production with the observed and the reduced
concentration of the precursor X, respectively.



## 3 Results and discussion

### 3.1 Overview of the metrological conditions and air pollutants concentrations

The time series of the hourly inorganic air pollutants, meteorological parameters, and TVOC concentrations are shown in Fig. 1. The discontinuity of the time series data is due to the failure of the instruments. The measured data from July to August 2018, September to November 2018, December 2018 to January 2019, and April to May 2020 are termed as summer, autumn, winter, and springtime data, respectively. Overall, the observed temperature and solar radiation gradually decreased from summer to winter and increased back to the summertime level in spring. The temperature ranged between -5.7 and 41.4 °C during the measurement period. The relative humidity values varied from 18 to 100% and high values were generally observed in winter and autumn. During the observation period, wind speed ranged between 0.1 and 7.5 ms$^{-1}$. Wind prevailed at the sampling site from many directions during the measurement periods; more details about the wind direction will be discussed in Sect.3.3.2. The $O_3$ and NOx concentrations varied from 2 to 160 ppbv and 0.4 to 90 ppbv, respectively. Whereas high $O_3$ concentrations (>80 ppbv) were observed in summer and spring, high NOx concentrations were measured in winter and at the end of autumn. The CO and $SO_2$ concentrations ranged from 83 to 3398 ppbv and 0.5 to 21 ppbv, respectively. Generally, high concentrations of CO and $SO_2$ were observed in winter and spring. The measured NO and $NO_2$ concentrations varied from 0.4 to 51 ppbv and 1 to 79 ppbv, respectively. In general, the high NO and $NO_2$ concentrations were observed in autumn and winter. The TVOCs concentrations varied between 9 and 393 ppbv during the observation period and the high values were measured in autumn and winter. More details about the abovementioned parameters will be discussed in the following section.

### 3.2 Concentration and composition of VOCs

In total 100 VOCs were observed in Nanjing industrial area, including 27 alkanes, 11 alkenes, 1 alkyne, 17 aromatics, 31 halohydrocarbons, 12 OVOCs, and 1 other (carbon disulfide) (Table S2). Ethane (5.8±2.5 ppbv), propane (4.2±1.5 ppbv), and ethylene (3±1.6 ppbv) were the most abundant VOCs in the study area during the observation period. However, we observed season-wise variations in the order of abundant VOC species (Table S2). For instance, acetone was the 3$^{rd}$ highest concentrated VOC in




spring. The abovementioned 4 VOC species are also frequently mentioned as the most abundant VOCs
in different regions in China (Deng et al., 2019; He et al., 2019; J. Li et al., 2018; Ma et al., 2019). We
compared the individual VOC concentrations with the available data presented in recent investigations.
The individual VOC concentrations in the current observation were similar to those in the previous
investigations in the same study area, however, they were almost twice of those found in a nonindustrial
suburban area in Nanjing (Table S2). The individual VOC concentrations in the current observation
were about 1.4 fold lower than those in Beijing (Li et al., 2015) and Shanghai (Zhang et al., 2018), but,
similar to those measured in Guangzhou (Zou et al., 2015). During the observation period, the
concentrations of different VOC-groups were in the order of alkanes (21±7 ppbv, 35%)>
halohydrocarbons (14.3±7.3 ppbv, 24%)> aromatics (9.9±5.8 ppbv, 17%)> OVOCs (7.5±1.9 ppbv,
13%)> alkenes (5±1.9 ppbv, 8%)> alkynes (1.4±0.3 ppbv, 2%)> others (0.5±0.2 ppbv, 1%). However,
we noticed relatively higher proportions of OVOCs (14% and 18%) than the aromatics (12% and 14%)
in summer and spring (Fig. 2c & f). The relatively higher contribution of OVOCs in summer and spring
could be related to the biogenic emissions (e.g. acetone, MEK from trees). Indeed, the relative
contribution of acetone and MEK to the TVOCs were higher in summer and spring than those in
autumn and winter (Table S2). Huang et al. (2019) reported that the industries, biogenic emissions, and
secondary formation are the main source of OVOCs in southern China. Relatively high proportions of
helohydrocarbons and aromatics were observed in autumn (25.7 and 19.3%, respectively) and winter
(25.8 and 17.6%, respectively) compared to those measured in summer (20.4 and 11.8%, respectively)
and spring (20.3 and 13.6%, respectively) (Fig. 2f). The high proportions of helohydrocarbons and
aromatics in autumn and winter could be related to the burning of biomass and fossil fuel for additional
heating. Similar to the observation in the current study, the alkane is generally the most abundant VOC
group in China (Mozaffar & Zhang, 2020). The relatively high contribution from halohydrocarbons to
the TVOCs could be related to the industrial emissions in the study area. In previous studies in an iron
smelt plant in Liaoning, China, a high concentration of halohydrocarbons (49%) was observed (Shi et
al., 2015). However, halohydrocarbons were not measured in previous investigations in the same study
area (An et al., 2014; An et al., 2017; Shao et al., 2016) and also in another suburban area in Nanjing
(Wu et al., 2020). Either aromatics or alkenes was mentioned as the second most abundant VOC-group



in those studies in Nanjing, which is the 3<sup>rd</sup> and 5<sup>th</sup> most abundant VOC group in the current investigation. In Shanghai, a nearby city, alkanes (42%) and alkenes (26%) were two major VOC-groups (Zhang et al., 2018). The TVOCs concentration was 59.8±28.6 ppbv over the whole observation period, and relatively higher TVOCs concentrations were measured in autumn (83±20 ppbv) and winter (77.5±16.8 ppbv) compared to those in spring (39.6±13.1 ppbv) and summer (38.8±10.2 ppbv). About 1.5-times higher TVOCs concentration was observed relative to the previous investigation in the same study area (An et al., 2014; An et al., 2017). Besides, we also found 3-times higher TVOCs concentration compared to the one in a nonindustrial suburban area in Nanjing (Wu et al., 2020). Halohydrocarbons and OVOCs were not measured in those previous studies in Nanjing, it could be one of the reasons for the relatively high TVOCs concentration in the current study. Observed autumn and wintertime TVOCs concentrations were similar to those measured in urban Beijing (86.2 ppbv in autumn) and Shanghai (94.1 ppbv in winter) (Li et al., 2015; Zhang et al., 2018). Similarly, observed summertime TVOCs concentration was similar to those found in urban Xi'an (42.6 ppbv), Wuhan (43.9 ppbv) (Zeng et al., 2018; Sun et al., 2019). However, yearly TVOCs concentration was 1.5-3 folds higher than those in Lanzhou, Wuhan, Tianjin, Ningbo, Chengdu, London, Los Angeles, and Tokyo (Jia et al., 2016; Hui et al., 2018; B. Liu et al., 2016a; Mo et al., 2017; Song et al., 2018; von Schneidemesser et al., 2010; Warneke et al., 2012; Hoshi et al., 2008). The diurnal variation of the TVOCs, alkenes, aromatics, halohydrocarbons, OVOCs, and alkanes concentrations showed a double-hump structure (Fig. 2a, b, d, & e). This double-hump pattern indicates the contribution of traffic emission during the rush-hours in the morning and evening. The lowest concentration of the TVOCs and different VOC-groups reached 12:00-16:00. Oppositely, the highest concentration of $O_3$ reached at that period (Fig. 3). The lowest $O_3$ concentrations were observed in winter which was consistent with the solar radiations.

## 3.3 Sources of VOCs

### 3.3.1 Specific Ratios

The use of the toluene/benzene (T/B) ratio is one of the simplest ways to preliminary analyse the VOC sources. If the T/B ratio is < 2, the study area is mainly affected by vehicle emissions (Hui et al., 2018,





2019). If the T/B ratio is > 2, the study area is influenced by other sources (e.g. industry, solvent utilization) beside vehicle emissions (Kumar et al., 2018; Niu et al., 2012; Li et al., 2019). Moreover, the T/B ratios are ranged between 0.2-0.6 in coal and biomass burning affected areas (Wang et al., 2009; Akagi et al., 2011). The diurnal variations in T/B ratios during different seasons are depicted in Fig. 4 (a, b, c, & d). The mean values of T/B ratios were ranged between 0.9-2 (1.4±0.3), 1.3-2 (1.7±0.2), 1.1-1.6 (1.4±0.1), and 1.4-2.7 (1.9±0.3) during summer, autumn, winter, and spring, respectively. As the mean values of T/B ratios were around 2, the study area could be mainly affected by vehicle emissions. The double-hump pattern in the diurnal variations in T/B ratios also indicates that the rush-hour traffic had a significant influence on the VOCs concentrations in the study area. Besides, the $75^{th}$ percentiles of T/B ratios were above 2 most of the investigation periods, therefore, the study area could also be influenced by industrial emissions.

Figure 4 (e, f, g, & h) shows the ratios of different alkanes and aromatics to acetylene. Acetylene is a tracer of combustion sources, the ratios of different alkanes and aromatics to acetylene are used to comprehend the contribution of other sources to combustion sources. The mean ratios of propane, n-butane, and i-butane to acetylene were around 2. 0-4.0, 0.7-1.6, and 0.4-0.8, respectively during all the seasons, which were smaller than those (11.5, 1.8, and 2.6, respectively) observed in Guangzhou city centre, which was affected by liquefied petroleum gas (LPG) emissions (Zhang et al., 2013). Therefore, LPG usages probably contributed a little fraction to the alkanes in the study area. The mean ratios of benzene, toluene, C8-aromatics, and C9-aromatics to acetylene were around 0.3-1. 0, 0.4-1.1, 0.2-0.6, and 0.1, respectively during all the seasons. The observed ratios of benzene and toluene to acetylene were much higher than those found in Jianfeng Mountains in Hainan (0.2 and 0.1, respectively) but comparable to those measured in urban Guangzhou (0.4 and 0.4-1, respectively) (Tang et al., 2007). Besides, the observed ratios of C8-aromatics and C9-aromatics to acetylene were comparable to traffic emission influenced urban Guangzhou (0.68 and 0.2, respectively) and Wuhan (0.5 and 0.2, respectively) (Zhang et al., 2013; Hui et al., 2018). Therefore, vehicle exhaust probably contributed significantly to the aromatics in the study area.



### 3.3.2 Potential Source Contribution Function (PSCF)

Besides the local sources, both the long and short distance transport of air mass could bring VOCs to the study area. Figure 5 shows the wind cluster and PSCF analysis results for different seasons. During summer, the major air masses were short-distance transports from the southwest (40%) and southeast (39%) directions. A minor air mass (21%) was transported from the east direction. High PSCF values were in the nearby southwest, southeast, and east directions; therefore, VOC pollution in the study area

was mainly affected by the short-distance transport from the south and east directions. During autumn, the dominant air masses were short-distance transport from the northeast (59%) and northwest (30%) directions. However, according to the PSCF analysis, VOC pollution was mainly influenced by the short distance transport from the south and east directions. During winter, short-distance transports from the northeast (46%) and northwest (37%) directions were the major incoming air masses to the study

area. According to the PSCF values, the short-distance air masses from the south and east directions were mainly transported VOC to the receptor site. During spring, air mass was mainly transported from the southwestern (49%) and eastern (30%) directions. Atmospheric VOCs to the study area were mainly transported by these two air masses mostly from the nearby areas. Overall, the high PSCF values were concentrated around the measurement site, therefore, short distance transports from the

surrounding areas and cities were the main reason for the high VOC concentration. The above conclusion perfectly makes sense as the sampling site is surrounded by different chemical and petrochemical industries, steel plants, gas stations, high traffic roads, and residential areas.

### 3.3.3 PMF Model Analysis

Differences were observed among the source profiles of VOCs obtained for different seasons (Sect. S1).

For instance, the biogenic source was identified in summer, biomass burning source was distinguished in autumn, and LPG/NG usage source was found in winter and spring. However, industry and vehicle-related VOC sources were identified during all the measurement seasons. According to PMF model results, aromatics were emitted from solvent usages, vehicle, and industry-related sources. Besides, industry and combustion processes were the main sources of halohydrocarbons and OVOCs. Moreover,

alkanes and alkenes were emitted from vehicle exhaust and fuel usage sources.



Figure 6 shows the relative contributions of different sources to ambient VOCs during different seasons. Overall, industry-related sources contributed to the major portion of the ambient VOC concentrations followed by vehicle emission. Industrial emission accounted for about 32%, 47%, 45%, and 23% in
summer, autumn, winter, and spring, respectively. The contributions of vehicle emission were about 34%, 26%, 24%, and 27% in summer, autumn, winter, and spring, respectively. The contribution of vehicle emission remained similar during the 4 seasons, however, the contribution of the industrial emission increased in autumn and winter. Previous investigations performed in Beijing, Tianjin, Wuhan, Chengdu, and Shuozhou also found that the industry and vehicle are the two most important
VOC sources (Zhang et al. 2017; Liu et al. 2016; Hui et al. 2018; Song et al. 2018) Jia et al., 2016). Besides these two sources, solvent usage (11%, 10%, 10%, and 4%, respectively) and gasoline evaporation (17%, 10%, NA, 6%, respectively) were two important VOC sources during those 4 seasons.  Moreover, source contribution from the biogenic source in summer (7%), biomass burning in autumn (7%), LPG/NG usage in winter (11%) and spring (18%), and multiple sources in winter (10%)
and spring (23%) was observed.

## 3.4 Chemical reactivity ($L^{OH}$) and contribution to $O_3$ and SOA formation

The estimated loss rates of OH radical ($L^{OH}$) with VOCs were about 2-fold high in autumn (13.7 s$^{-1}$) and winter (13.5 s$^{-1}$) compared to those in summer (7 s$^{-1}$) and spring (7.5 s$^{-1}$) (Fig. 7 a). The relatively high $L^{OH}$ values in autumn and winter were due to the relatively high VOC concentrations in those
340 seasons (Fig.2). The average $L^{OH}$ value was 10.4±3.6 s$^{-1}$ over the four seasons. It was in a similar range with the values determined in Guangzhou (10.9 s$^{-1}$), Chongqing (10 s$^{-1}$), Xian (1.6-16.2 s$^{-1}$), and Tokyo (7.7-13.4 s$^{-1}$), however, higher than the values estimated in Shanghai (2.9-5 s$^{-1}$, 6.2 s$^{-1}$) and Beijing (7 s$^{-1}$) (Tan et al., 2019; Zhu et al., 2019; Yoshino et al., 2012; Song et al., 2020). While alkene was the highest contributor to the $L^{OH}$ in summer (3 s$^{-1}$, 43%) and spring (2.6 s$^{-1}$, 35%), aromatic was the
345 maximum contributor in autumn (6.9 s$^{-1}$, 50%) and winter (5.9 s$^{-1}$, 44%)  (Fig. 11 a & d). An increase in the OH loss rate by OVOCs was observed in spring (17%) compared to the other seasons (10, 8, and 9% in summer, autumn, and winter, respectively). Over the four seasons, the contribution of VOC-groups to



$L^{OH}$ exhibited the following trend: aromatics > alkenes > alkanes > OVOCs > halohydrocarbons. Similar to the current study, aromatic is also mentioned as the maximum contributors to $L^{OH}$ in different

regions in China, however, the alkene is generally reported as the top contributor to $L^{OH}$ (Zhang et al., 2020; Zhao et al., 2020; Hui et al., 2018; Song et al., 2020). Figure 7 also shows the top 10 VOCs contributing to $L^{OH}$ for different seasons. Whereas isoprene was the highest contributor to $L^{OH}$ in summer, styrene was the largest contributor in autumn and winter. On the other hand, naphthalene was the main contributor to $L^{OH}$ in spring. Overall, styrene, naphthalene, ethylene, and isoprene were the

main contributor to $L^{OH}$ in the study area. In previous studies in China, these compounds are also mentioned as one of the highest contributors to $L^{OH}$ (Zhao et al., 2020; Hui et al., 2018; Song et al., 2020).

The estimated $O_3$ formation potential (OFP) of VOCs were about 2-times high in autumn (170.8 ppbv) and winter (175.4 ppbv) relative to those in summer (86.2 ppbv) and spring (82.8 ppbv) (Fig. 8 a). The

average OFP value was 128.8±51.2 ppbv during the measurement period. The springtime OFP was similar to the one estimated in Beijing (80 ppbv) (Li et al., 2015). The summertime OFP was about 1.5 times higher than the one in Xi'an (Song et al., 2020), but, about 1.4-2 folds lower than those found in Shanghai (Liu et al., 2019). The average OFP was about 1.5 times higher than the one in Wuhan (Hui et al., 2018). Whereas alkene was the major contributor to OFP in summer (37.4 ppbv, 43%), winter

(72.8 ppbv, 41%), and spring (31.6 ppbv, 38%), aromatics contributed the most to OFP in autumn (62.7 ppbv, 37%) (Fig. 12 a & d). During the measurement period, the contribution of VOC-groups to OFP showed the following trend: alkenes > aromatics > alkanes > OVOCs > halohydrocarbons. The alkene is also mentioned as the top contributor to OFP in Nanjing and the same observation is commonly found in China (An et al., 2014; Hui et al., 2018; Song et al., 2018; Song et al., 2020). The top 10 VOCs

contributing to OFP for different seasons are also shown in Fig. 8 (b, c, e, & f). Ethylene was the major contributor to OFP during all the season. Followed by ethylene, cis-1,3-dichloropropene was the main contributor to OFP from summer to winter. In spring, propylene was the second most contributors to OFP. Overall, different alkenes were the highest contributor to OFP in the study area. Alkenes are also mentioned as the top contributor to OFP in the previous investigations in Nanjing (An et al., 2014).



Therefore, the reduction of these alkenes emissions in the study area could be one of the ways to reduce ambient $O_3$ concentration.

The secondary organic aerosol potentials (SOAP) were about 3-times higher in autumn (1422 ppbv) and winter (1269 ppbv) than those in summer (466 ppbv) and spring (398 ppbv) (Fig. 9a). The average SOAP was 889±531 ppbv during the measurement period. The average SOAP was about 2-3 times

380 higher than those estimated in Wuhan and Beijing (Hui et al., 2019; Li et al., 2020). Aromatics was the main contributor to SOAP during all the seasons (95-97%) (Fig. 9 a & d) which was consistent with the observations in Chengdu (Song et al., 2018), Beijing (Li et al., 2020), and Wuhan (Hui et al., 2019). During the measurement period, the contribution of VOC-groups to SOAP exhibited the following trend:  aromatics > alkanes > alkenes > OVOCs. Styrene, cumene, toluene, benzene, and o-xylene were

385 the major contributor to SOAP during all the season (Fig. 9 b, c, e, & f). Therefore, the reduction of these aromatics emissions in the study area could be one of the ways to reduce ambient SOA concentration.

### 3.5 Sensitivity analysis of $O_3$ formation

Figure 10 shows the EKMA isopleth diagrams of $O_3$ for different seasons. In all the diagrams, VOC and

390 NOx = 100 % is the base case. The ridgeline divided the diagrams into two regimes, VOC-sensitive (above) and NOx-sensitive (below) regimes. For all the seasons, the study area fell above the ridgeline. Moreover, a decrease in $O_3$ production was noticed with the decrease in VOC concentration. Therefore, the study area was in the VOC-sensitive regime for $O_3$ formation during all the seasons. As a case study, $O_3$ formation sensitivity to its precursors was tested on a high $O_3$ concentration day (July 29

2018, maximum 126 ppbv). During the high $O_3$ episode, the study area was also in the VOC-sensitive regime for $O_3$ formation (Fig. S5). We also employed the RIR analysis to evaluate the $O_3$ production sensitivity to VOC, NOx, and CO concentrations (Fig. 11). The RIR value of VOC was the highest during all the seasons. It indicates that the $O_3$ production was more sensitive to the reduction of VOC concentration. This finding is consistent with the above results in the EKMA isopleth (Fig. 10).  Except

for the spring, the RIR values of CO were very small relative to those for the VOC. It indicates that the CO concentrations were relatively less important for the $O_3$ formation during those seasons. The RIR





values for NOx were negative during all the seasons, implying that the O$_3$ formation was in the NOx-titration regime in the study area. From the above analysis, it is evident that a reduction of VOC concentration in the study area will be the most efficient way to reduce the O$_3$ formation. The previous two studies performed in Nanjing also concluded the same finding based on VOC/NOx ratios and RIR analysis (An et al., 2015; Xu et al., 2017). Our findings are also consistent with the previous studies performed in other regions in China (Tan et al., 2018a; He et al., 2019; Feng et al., 2019; Ma et al., 2019). However, NOx-sensitive regions for O$_3$ formation are also found in China (Tan et al., 2018; Jia et al., 2016).

## 4 Conclusions

Industries are an important anthropogenic source of VOCs. VOC plays a major role in tropospheric chemistry and air quality. Nanjing is one of the biggest industrial cities in China. We performed a long term investigation of ambient VOCs in an industrial area in Nanjing. About 1.5 and 3-folds high TVOCs concentrations were observed compared to those previously reported in the same study area and a nonindustrial suburban area in Nanjing, respectively. The relatively high TVOCs was due to halohydrocarbons and OVOCs concentrations were not measured in those previous studies in Nanjing. Therefore, halohydrocarbons and OVOCs were an important part of the TVOCs in Nanjing, and industrial emissions had a large influence on VOC concentration in the study area. Observed TVOCs concentration was also about 1.5-3 folds higher than those reported in other cities in China and the world, but, similar to those measured in urban Beijing and Shanghai. This high VOC concentration in the study area needs to be reduced to decrease O$_3$ concentration and improve the local air quality. TVOCs concentrations were about 2-times high in autumn and winter compared to those in summer and spring. Generally, haze pollutions frequently happen in autumn and winter, therefore, VOC concentration reduction in these seasons is an important step to reduce haze pollutions in the study area. After alkane, halohydrocarbon was the 2$^{nd}$ largest contributor to the TVOCs, indicating a high influence of industrial emissions. Generally, alkenes/aromatics/OVOCs are the 2$^{nd}$ largest contributor to the TVOCs in China, therefore, industries in Nanjing emitted a high amount of halohydrocarbons into the atmosphere. As halohydrocarbons are carcinogenic, their emissions should be reduced. PSCF analysis



indicated that the short distance transports from the surrounding areas and cities were the main reason for high VOC concentration in the study area. Hence, local emissions should be reduced to decrease the haze and $O_3$ pollution in the study area. Industries were the major VOC sources in the study area followed by vehicles, thus, emission reduction from these two sources should get more priority. Aromatics and alkenes accounted for most of the $L^{OH}$, OFP, and SOAP, thus, these 2 kinds of VOCs should get more priority in emission reduction policies and strategies. During all the seasons, the study area was in the VOC-sensitive regime for $O_3$ formation. Therefore, VOCs especially aromatics and alkenes emission reduction is the most effective way to decrease the local $O_3$ formation.

**Data availability**

All the data presented in this article can be accessed through https://osf.io/bm6cs/.

**Author contribution**

YLZ designed and supervised the project; MYF, FX, YCL, FC, and AM conducted the measurements; AM analysed the data and prepared the manuscript. All authors contributed in discussion to improve the article.

**Competing interests**

The authors declare that they have no conflict of interest.

**Acknowledgements**

The authors thank funding support from the National Nature Science Foundation of China (No. 41977305 and 41761144056), the Provincial Natural Science Foundation of Jiangsu (No. BK20180040), and the Jiangsu Innovation & Entrepreneurship Team. We are also grateful to Zijin Zhang and Meng-Yao Cao for their help on sampling.

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





**Figure 1: Time series of hourly meteorological parameters, inorganic air pollutants, and TVOCs concentrations during the observation period at Nanjing. The green, yellow, cyan, and light-green shaded areas indicate summer, autumn, winter, and spring seasons, respectively. The discontinuity of the measured data is due to the instruments failure.**





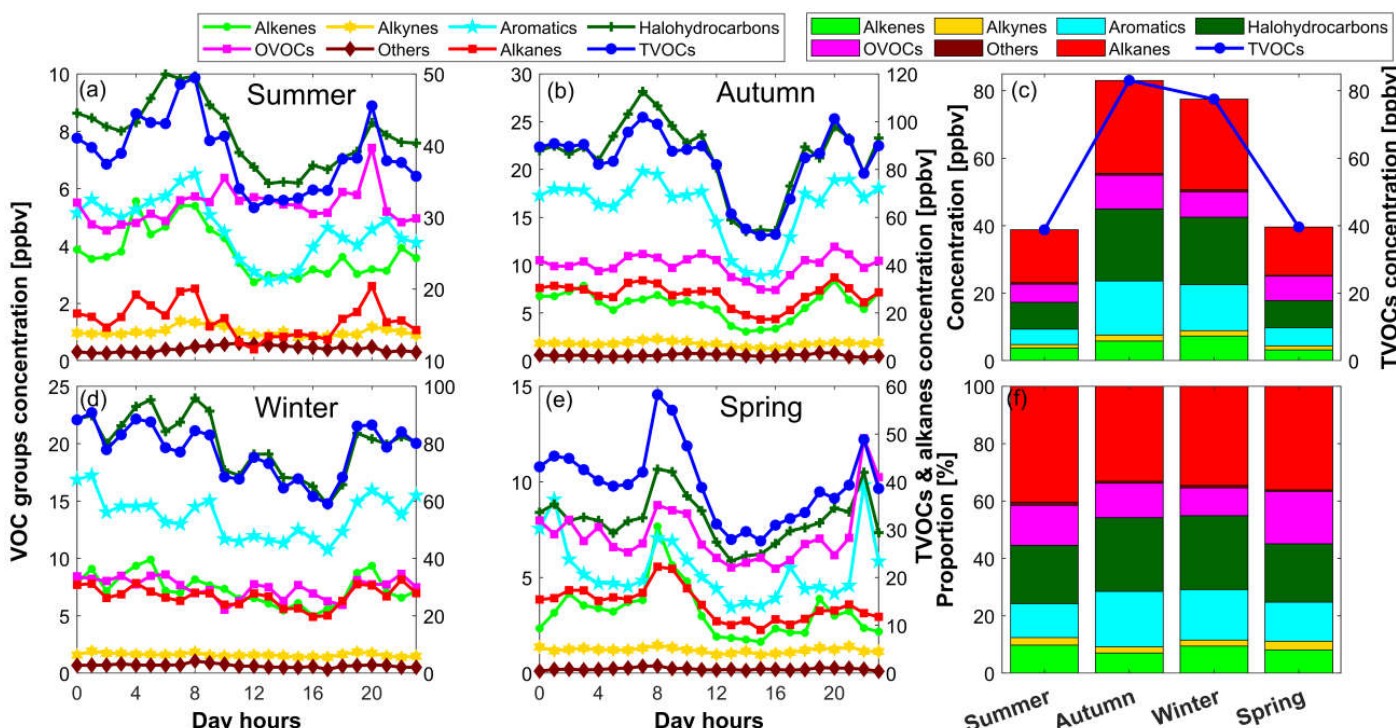

**Figure 2: Diurnal variations in TVOCs and different VOC-groups concentrations in different seasons (a, b, d, & e) and seasonal variations in average concentrations and proportion of TVOCs and different VOC-groups (c & f).**





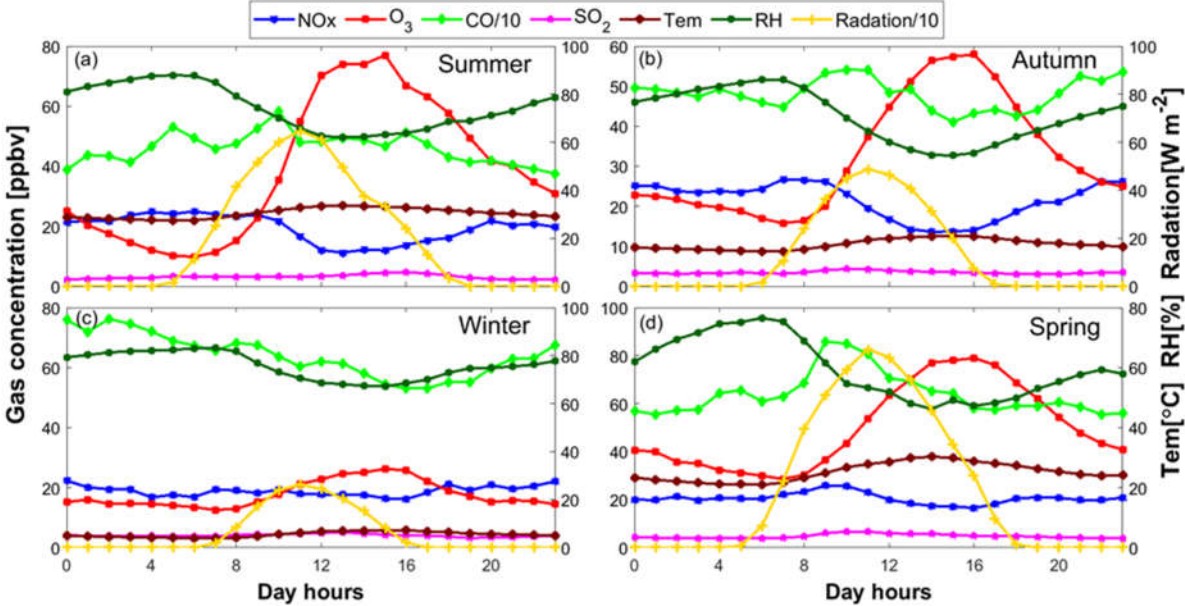

**Figure 3: Diurnal variations in weather conditions and NOx, O3, CO, and SO2 concentrations in different seasons. Note that the plotted CO concentrations and solar radiation values are reduced by 10-folds for a better visualization.**



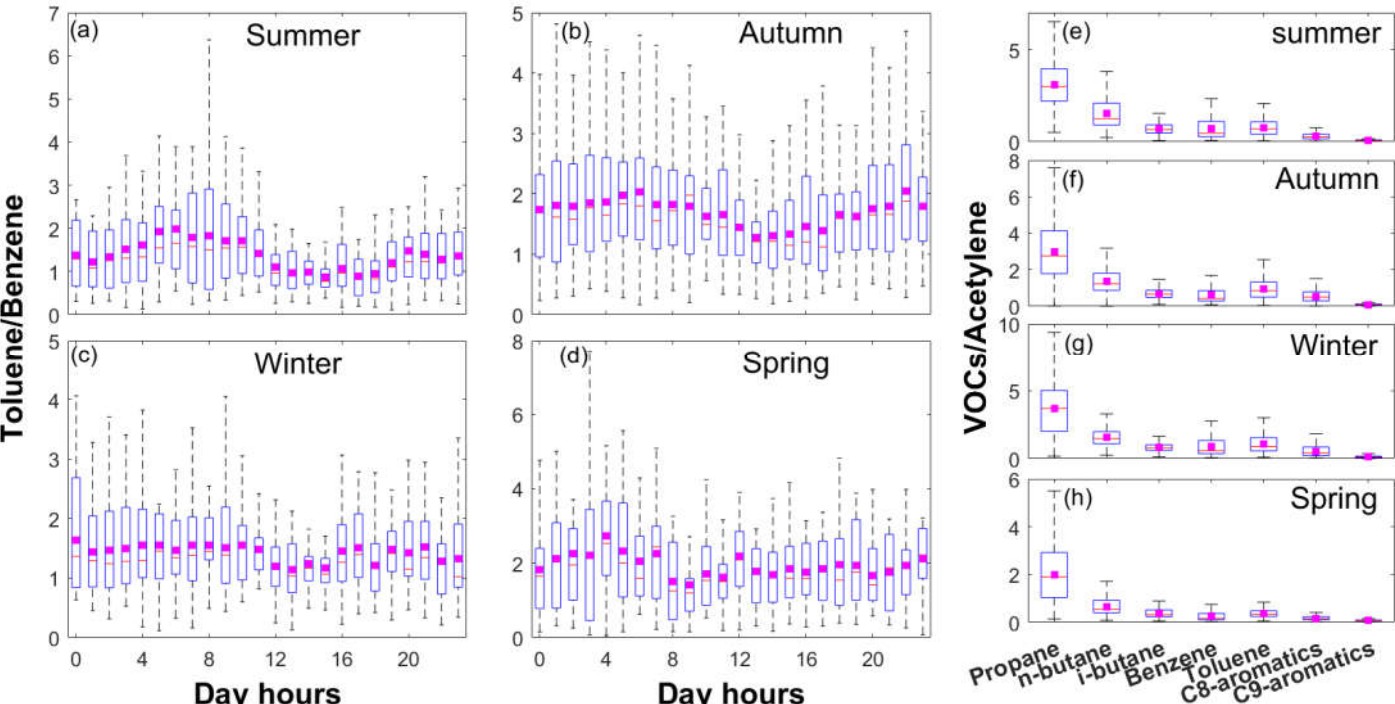

**Figure 4: Diurnal variations in toluene/benzene ratios (a, b, c, & d) and in the ratios of different VOCs to acetylene in different seasons (e, f, g, & h). The pink-colored squares in the box-plots represent the average values.**





**Figure 5: Wind cluster and PSCF analysis during (a) summer (b) autumn, (c) winter, and (d) spring based on the 24 hours backward air mass trajectories from the study area.**



**Figure 6: relative contributions of different sources to ambient VOCs in Nanjing industrial area during different seasons**





**Figure 7: Contribution to OH loss rates of different VOC-groups and the top 10 VOC species in different seasons**





**Figure 8: Contribution to ozone formation potential of different VOC-groups and the top 10 VOC species in different seasons**





**Figure 9: Contribution to secondary organic aerosol formation potential of different VOC-groups and the top 10 VOC species in different seasons**

**Figure 10: O₃ isopleth diagram for (a) summer (b) autumn, (c) winter, and (d) spring based on percentage changes in VOCs and NOx concentrations in Nanjing and corresponding modelled O₃ production.**





**Figure 11: The RIR values of the VOC, NOx, and CO for the different seasons in Nanjing**