# Peer review of "Measurement report: High Contributions of Halohydrocarbon and Aromatic Compounds to Emissions and Chemistry of Atmospheric VOCs in Industrial Area"

_Atmospheric Chemistry and Physics, 2021_

## Referee Comment (RC3)

This manuscript presents VOC measurements in Nanjing China and corresponding PMF, PCSF, OFP, SOAP, EKMA, RIR, tracer/tracer ratio, and LOH analyses. The primary improvement over previous investigations in this area are (1) additional inclusion of halocarbons and a limited set of OVOCs (2) a multi-season dataset. While the dataset seems valuable and list of analysis techniques is exhaustive, it is not clear what substantial new results are presented. As discussed below, some analysis methods and/or their results are not well-described.

1. PMF analysis-

- PMF was performed separately on each season. Was there enough data in during 2020 to draw meaningful conclusions? It is difficult to tell the duration of measurements from figure 1.
- Do the PMF results indicate that there is seasonality in source sector emissions, or are the fingerprints consistent?
- Why was an 8 factor solution chosen (v. 7 v. 6 factors, etc)?
- The supplemental figures are difficult to read due to the number of species included in the figure. It should be noted which axis should be read with each dataset.
- As noted by reviewer 2, the results are odd (e.g., biogenic isopentane; propane from solvents). The revised and trimmed-down PMF submitted in response to the reviewer still has these oddities.
- 2. PSCF analysis-
  - I do not think this method is ubiquitous enough to merit the brevity of explanation. For example, what do the colors mean in figure 5? What does the impact of a 24 v. 48 v.
    72 h back-trajectory have on the analysis? Does this assume that VOCs have the same lifetime as the back-trajectory?
  - It is not clear what useful/new information is derived from this analysis. The main results of this analysis (line 428-430) seems to be that the VOCs measured at the site come primarily come from industries located near the site. Would there be a reason to think otherwise?

**3. OFP/EKMA/RIR**

- Discussion of wintertime OFP seems unusual, as photochemical ozone production in the winter is not a primary concern.
- There are not enough details in the description of the F0AM model. For example: Which VOCs were constrained (not all are in MCM)?

Other comments-

- I find the title "High Contributions of Halohydrocarbon and Aromatic Compounds to Emissions and Chemistry of Atmospheric VOCs in Industrial Area" to be misleading. Halocarbons do not contribute significantly to either SOA or ozone chemistry. Neither halocarbons nor aromatics dominate calculated OFP in the summer (Figure 5).
- It seems a major conclusion (line 431) is that industries and vehicles should have the most priority in reducing emissions. Nearly all anthropogenic VOCs in this area could

be classified as industry/vehicles, so this conclusion seems too vague to be meaningful. Is it a new finding?

- It is unusual to see the notation KOH (typically lowercase/supscripts are for kinetics, uppercase for equilibrium reactions).
- There are no units in supplemental tables.
- It is not clear how to interpret numbers given in lines 272 (e.g., 0.9-2 (1.4 +/- 0.3))

---

## Author Response (AR1)

**Authors' response to referee 1**

We would like to thank the anonymous Referee #1 who gave his/her time to provide the comments. First of all, the manuscript is a "measurement report" which presents substantial new results on atmospheric VOCs observed in one of the most important industrial areas in China. It is not a "Research articles" type manuscript. Secondly, the objective of the study/manuscript was not to report the process-specific/industry-specific VOC emissions, the objective was to present the characteristics of VOCs in an industry affected area. There have been many process-specific/industry-specific VOC emissions studies already performed (e.g. Mo et al., 2015; Zhang et al., 2017; Yan et al., 2016; Shi et al., 2015).

**References**

Mo, Z., Shao, M., Lu, S., Qu, H., Zhou, M., Sun, J., & Gou, B. (2015). Process-specific emission characteristics of volatile organic compounds ( VOCs ) from petrochemical facilities in the Yangtze River Delta , China. *Science of the Total Environment*, *533*, 422–431. https://doi.org/10.1016/j.scitotenv.2015.06.089

Shi, J., Deng, H., Bai, Z., Kong, S., Wang, X., Hao, J., … Ning, P. (2015). Emission and profile characteristic of volatile organic compounds emitted from coke production, iron smelt, heating station and power plant in Liaoning Province, China. *Science of the Total Environment*, *515–516*(x), 101–108. https://doi.org/10.1016/j.scitotenv.2015.02.034

Yan, Y., Yang, C., Peng, L., Li, R., & Bai, H. (2016). Emission characteristics of volatile organic compounds from coal-, coal gangue-, and biomass-fired power plants in China. *Atmospheric Environment*, *143*, 261–269. https://doi.org/10.1016/j.atmosenv.2016.08.052

Zhang, Z., Wang, H., Chen, D., Li, Q., Thai, P., Gong, D., … Wang, B. (2017). Emission characteristics of volatile organic compounds and their secondary organic aerosol formation potentials from a petroleum re fi nery in Pearl River Delta , China. *Science of the Total Environment*, *584–585*, 1162–1174. https://doi.org/10.1016/j.scitotenv.2017.01.179

**Authors' response to referee 2**

The authors are grateful to the anonymous referee #2 who gave his/her time to provide helpful comments and suggestions that improved our manuscript. Here, we present the answers for each of the comments. The manuscript and supplement with tracked changes can be found at the end of the document.

- **The writing of the paper obscured the fact that the observations were not continuous. The dataset includes ~ 1 month in winter, spring, and summer and 3 months in autumn. Some of the differences in comparison to other studies may be due to the comparison of 1 month data to a season.**

Authors' response: The information about the investigation period is now clearly mentioned in

10    the abstract, introduction, and methods sections. Besides, the time series plot (Figure 1) also shows the observation period clearly and it is mentioned in section 3.1. Comparison with other studies is now done carefully. The observation periods of the previous studies are also mentioned in the text. Most of the previous studies we presented in our comparison discussion did not present any time series plot for the whole observation period and also did not mention clearly

15    that the measurement was continuously performed for a season or a year. So, it is not certain that we are comparing our 1-3 months data to a season.

- **I do not suggest including halocarbons in TVOC comparison. Halocarbons are not reactive and do not contribute significantly to OH reactive or the productions of ozone and SOA. There are good reasons for not including halocarbons in TVOC**

20    **data in previous studies. In the discussion of Line 250-260, it is unclear if and what halocarbons and OVOCs were included in TVOCs in previous studies. Not knowing that, the comparison results can be misleading.**

Authors' response: Many previous studies also included halocarbons in TVOC estimation and we already included those reported TVOC concentrations in our comparison discussion (e.g. Li

25    et al., 2015; Zhang et al., 2018; Zeng et al., 2018; Sun et al., 2019). We still want to compare our observation with these studies. That is why we are now reporting TVOC concentration with halocarbons and without halocarbons. Now we have rewritten the TVOC comparison part. We checked carefully each previous study to find out what kind of VOCs they included in their TVOC estimation and the comparison with our observation is done accordingly. Moreover,

30    OVOCs are also not included in TVOC estimation in several studies (e.g. Zhang et al., 2018; Jia

et al., 2016; Hui et al., 2018; Liu et al., 2016a; Mo et al., 2017; Song et al., 2018). That is why we also estimated TVOC without OVOCs and compare it with those studies.

- **The discussion of section 3.3.1 can be removed. T/B ratio is meaningful only if they are from a single source. The PMF results show that T and B are from multiple sources and T/B ratio is not meaning. The same argument applies to the ratios of alkanes or aromatics to acetylene.**

Authors' response: Section 3.3.1 is removed.

- **The PMF results are confusing. In summer, for example, ethane is from vehicle, isopentane is from biogenic sources, ethane and propane are from vehicles, and propane is from solvent use. None of these make much sense. It's more or less the same for other seasons. In autumn, biomass burning has ethane but not acetylene. In winter, vehicle 1 and 2 are similar. Spring and autumn have gasoline evaporation, but summer and winter do not. Gasoline evaporation should be largest in summer. LPG/NP does not have ethylene in spring but has most of ethylene in winter.**

Authors' response: PMF model analysis is carried out again using typical tracers of various emission sources and the text has been modified accordingly.

- **PSFC analysis is applicable only to VOCs that are short lived because the backtrajectories are only 24 hours.**

Authors' response: New PSCF analysis based on 72 hours backward air mass trajectories is now added. The text has been modified according to the new PSCF analysis.

- **I do not feel OFP and SOAP analyses are useful (although they are often included in VOC papers). Figures 7 and 8 show that VOCs have much higher OFP and SOAP in winter than summer, which is technically true but we know the contributions of VOCs to ozone and SOA are much larger in summer than winter.**

60   Authors' response: The OFP and SOAP analyses are removed.

- **The EKMA diagrams of Figure 10 show ozone at ~50 ppbv for 100% NOx and VOCs in summer. Ozone here is daily 1 hour maximum, I think. 50 ppbv seems low for daily 1 hour maximum in summer. One possible reason is that the Thermo**
65   **chemiluminescence instrument can severely overestimate NO2 because of the conversion of NOy to NOx.**

Response: Yes, the Thermo chemiluminescence instrument can overestimate $NO_2$ concentrations (Dunlea et al., 2007). However, there is also evidence that it does not significantly overestimate $NO_2$ concentrations (Chate et al., 2014). Anyway, following Dunlea et al. (2007), we ran the box
70   model using $NO_2$ concentrations reduced by 22% and made the summertime EKMA diagram (Figure R1 below). However, the 1-hour maximum $O_3$ concentration for 100% NOx and VOCs did not increase.

[Figure]

Figure R1: O$_3$ isopleth diagram for summer based on percentage changes in VOCs and NOx concentrations in Nanjing and corresponding modelled O$_3$ production. In the model input, the NO$_2$ concentrations were reduced by 22%.

- **"Halocarbons" was mis-spelled in several places.**

Authors' response: Amended. The term "halocarbon" is used everywhere instead of "halohydrocarbon".

References

Chate, D. M., Ghude, S. D., Beig, G., Mahajan, A. S., Jena, C., Srinivas, R., … Kumar, N. (2014). Deviations from the O3-NO-NO2 photo-stationary state in Delhi, India. *Atmospheric Environment*, *96*(2), 353–358. https://doi.org/10.1016/j.atmosenv.2014.07.054

Dunlea, E. J., Herndon, S. C., Nelson, D. D., Volkamer, R. M., San Martini, F., Sheehy, P. M., … Molina, M. J. (2007). Evaluation of nitrogen dioxide chemiluminescence monitors in a polluted urban environment. *Atmospheric Chemistry and Physics*, *7*(10), 2691–2704. https://doi.org/10.5194/acp-7-2691-2007

Hui, L., Liu, X., Tan, Q., Feng, M., An, J., Qu, Y., … Jiang, M. (2018). Characteristics, source apportionment and contribution of VOCs to ozone formation in Wuhan, Central China. *Atmospheric Environment*, *192*(2), 55–71. https://doi.org/10.1016/j.atmosenv.2018.08.042

Jia, C., Mao, X., Huang, T., Liang, X., Wang, Y., Shen, Y., … Gao, H. (2016). Non-methane hydrocarbons (NMHCs) and their contribution to ozone formation potential in a petrochemical industrialized city, Northwest China. *Atmospheric Research*, *169*, 225–236. https://doi.org/10.1016/j.atmosres.2015.10.006

Li, J., Xie, S. D., Zeng, L. M., Li, L. Y., Li, Y. Q., & Wu, R. R. (2015). Characterization of ambient volatile organic compounds and their sources in Beijing, before, during, and after Asia-Pacific Economic Cooperation China 2014. *Atmospheric Chemistry and Physics*, *15*(14), 7945–7959. https://doi.org/10.5194/acp-15-7945-2015

Liu, B., Liang, D., Yang, J., Dai, Q., Bi, X., Feng, Y., … Xu, H. (2016). Characterization and source apportionment of volatile organic compounds based on 1-year of observational data in Tianjin, China. *Environmental Pollution*, *218*, 757–769. https://doi.org/10.1016/j.envpol.2016.07.072

Mo, Z., Shao, M., Lu, S., Niu, H., Zhou, M., & Sun, J. (2017). Characterization of non-methane

105       hydrocarbons and their sources in an industrialized coastal city , Yangtze River Delta , China. *Science of the Total Environment*, *593–594*, 641–653. https://doi.org/10.1016/j.scitotenv.2017.03.123

Song, M., Tan, Q., Feng, M., Qu, Y., & Liu, X. (2018). Source Apportionment and Secondary Transformation of Atmospheric Nonmethane Hydrocarbons in Chengdu , Southwest China. *Journal of Geophysical Research Atmospheres*, *123*(2), 9741–9763. https://doi.org/10.1029/2018JD028479

Sun, J., Shen, Z., Zhang, Y., Zhang, Z., Zhang, Q., Zhang, T., … Li, X. (2019). Urban VOC profiles, possible sources, and its role in ozone formation for a summer campaign over Xi'an, China. *Environmental Science and Pollution Research*, *26*(27), 27769–27782. https://doi.org/10.1007/s11356-019-05950-0

Zeng, P., Lyu, X. P., Guo, H., Cheng, H. R., Jiang, F., Pan, W. Z., … Hu, Y. Q. (2018). Causes of ozone pollution in summer in Wuhan, Central China. *Environmental Pollution*, *241*(x), 852–861. https://doi.org/10.1016/j.envpol.2018.05.042

Zhang, Y., Li, R., Fu, H., Zhou, D., & Chen, J. (2018). Observation and analysis of atmospheric volatile organic compounds in a typical petrochemical area in Yangtze River. *Journal of Environmental Sciences*, *71*, 233–248. https://doi.org/10.1016/j.jes.2018.05.027

**Measurement report: High Contributions of Halocarbon and Aromatic Compounds to Emissions and Chemistry of Atmospheric VOCs in Industrial Area**

Ahsan Mozaffar [1,2,3], Yan-Lin Zhang [1,2,3*], Yu-Chi Lin[1,2,3], Feng Xie [1,2,3], Mei-Yi Fan [1,2,3], and Fang Cao [1,2,3]

[1]Yale-NUIST Center on Atmospheric Environment, International Joint Laboratory on Climate and Environment Change, Nanjing University of Information Science and Technology, Nanjing, 210044, China.

[2]Key Laboratory Meteorological Disaster; Ministry of Education & Collaborative Innovation Center on Forecast and Evaluation of Meteorological Disaster, Nanjing University of Information Science and Technology, Nanjing, 210044, China.

[3]Jiangsu Provincial Key Laboratory of Agricultural Meteorology, College of Applied Meteorology, Nanjing University of Information Science & Technology, Nanjing 210044, China.

*Correspondence to*: Yan-Lin Zhang (dryanlinzhang@outlook.com)

**Abstract.** Volatile organic compounds (VOCs) are key components for tropospheric chemistry and air quality. We investigated ambient VOCs in an industrial area in Nanjing, China for about 1 month in winter, spring, and summer and 3 months in autumn from between July 2018 to and May 2020. The total VOCs (TVOCs) concentration was 59.8±28.6 ppbv during the investigation period. About twice TVOCs concentrations were observed in autumn (83±20 ppbv) and winter (77.5±16.8 ppbv) seasons compared to those in spring (39.6±13.1 ppbv) and summer (38.8±10.2 ppbv). Unlike In previous studies in Nanjing, oxygenated-VOCs (OVOCs) and halocarbons were not measured, the observed TVOCs was about 1.5 and 3 times higher than those previously reported in the same study area and a nonindustrial suburban area in Nanjing, respectively the current TVOCs concentration without halocarbons and OVOCs was similar to the previous investigation in the same study area, however, 2 folds higher than the one reported in the nonindustrial suburban area in Nanjing. Observed TVOCs concentrations were was similar to those the one in the metropolitan city Beijing, and but, 1.3 fold smaller than the one reported in Shanghai., Observed TVOCs concentration was similar to those in Lanzhou, Chengdu, Tokyo, and Xi'an, 
[revised manuscript text omitted]

385 concentrations in the current observation were about 1.4 fold lower than those measured in Beijing during October-November (Li et al., 2015). The winter time individual VOC concentrations in the current observation were also about 1.4 fold lower than those measured in and Shanghai during November-January (Zhang et al., 2018)., butBut, the yearly individual VOC concentrations in the current observation were similar to those measured in Guangzhou during

390 June-May (Zou et al., 2015). During the observation period, the concentrations of different VOC-groups were in the order of alkanes (21±7 ppbv, 35%)> halocarbons (14.3±7.3 ppbv, 24%)> aromatics (9.9±5.8 ppbv, 17%)> OVOCs (7.5±1.9 ppbv, 13%)> alkenes (5±1.9 ppbv, 8%)> alkynes (1.4±0.3 ppbv, 2%)> others (0.5±0.2 ppbv, 1%). However, we noticed relatively higher proportions of OVOCs (14% and 18%) than the aromatics (12% and 14%) in summer

395 and spring (Fig. 2c & f). The relatively higher contribution of OVOCs in summer and spring could be related to the biogenic emissions (e.g. acetone, MEK from trees). Indeed, the relative contribution of acetone and MEK to the TVOCs were higher in summer and spring than those in autumn and winter (Table S2). Huang et al. (2019) reported that the industries, biogenic emissions, and secondary formation are the main source of OVOCs in southern China. Relatively

400 high proportions of healohydrocarbons and aromatics were observed in autumn (25.7 and 19.3%,

respectively) and winter (25.8 and 17.6%, respectively) compared to those measured in summer (20.4 and 11.8%, respectively) and spring (20.3 and 13.6%, respectively) (Fig. 2f). The high proportions of h~ae~lo~hydro~carbons and aromatics in autumn and winter could be related to the burning of biomass and fossil fuel for additional heating. Similar to the observation in the current study, the alkane is generally the most abundant VOC group in China (Mozaffar & Zhang, 2020). The relatively high contribution from halocarbons to the TVOCs could be related to the industrial emissions in the study area. In previous studies in an iron smelt plant in Liaoning, China, a high concentration of halocarbons (49%) was observed (Shi et al., 2015). However, halocarbons and OVOCs were not measured in previous investigations in the same study area (An et al., 2014; An et al., 2017; Shao et al., 2016) and also in another suburban area in Nanjing (Wu et al., 2020). Either aromatics or alkenes was mentioned as the second most abundant VOC-group in those studies in Nanjing, which is the 3rd and 5th most abundant VOC group in the current investigation. In Shanghai, a nearby city, alkanes (42%) and alkenes (26%) were two major VOC-groups and halocarbons and OVOCs were not reported (Zhang et al., 2018). The TVOCs concentration including halocarbons was 59.8±28.6 ppbv over the whole observation period, and relatively higher TVOCs concentrations were measured in autumn (83±20 ppbv) and winter (77.5±16.8 ppbv) compared to those in spring (39.6±13.1 ppbv) and summer (38.8±10.2 ppbv). The TVOCs concentration without halocarbons were 45.4±20.4, 61.7±14.6, 57.4±11.8, 31.6±10.9, and 30.9±8.2 ppbv during the whole observation period, autumn, winter, spring and summer, respectively. ~About 1.5-times higher TVOCs concentration was observed relative to the previous investigation in the same study area (An et al., 2014; An et al., 2017). Besides, we also found 3-times higher TVOCs concentration compared to the one in a nonindustrial suburban area in Nanjing (Wu et al., 2020).~ As mentioned before, ~H~halocarbons and OVOCs were not ~measured~ reported in the previous investigation in the same study area (An et al., 2014; An et al., 2017) and in a nonindustrial suburban area in Nanjing (Wu et al., 2020). The current TVOCs concentration without halocarbons and OVOCs was similar to the previous investigation in the same study area, however, 2 folds higher than the one reported in the nonindustrial suburban area in Nanjing ~those previous studies in Nanjing, it could be one of the reasons for the relatively high TVOCs concentration in the current study~. Observed autumn~-time~ ~and wintertime~ TVOCs concentrations including halocarbons ~were~ was similar to ~those~ the one measured in urban Beijing (86.2 ppbv ~in autumn~during 17-31 October) (Li et al., 2015). ~and~ Wintertime TVOCs

concentrations without OVOCs in Shanghai (94.1 ppbv in winterNov-Jan) was higher than the current observation (70±15.1 ppbv)  (Zhang et al., 2018). Observed summertime TVOCs concentration was similar to those found in urban Xi'an (42.6 ppbv, estimation includes halocarbons and OVOCs), Wuhan (43.9 ppbv estimation includes halocarbons and OVOCs) (Zeng et al., 2018; Sun et al., 2019). Yearly TVOCs concentration including halocarbons and OVOCs was 1.7 folds higher than the one found in Wuhan (Hui et al., 2018b). Besides, yearly TVOCs concentration without halocarbons and OVOCs (37.9 ppbv) was similar to Lanzhou, Chengdu, and Tokyo, however, 1.5 times higher than those reported for Tianjin and Ningbo (Jia et al., 2016;(Hui et al., 2018b;) B. Liu et al., 2016a; Hoshi et al., 2008; Mo et al., 2017; Song et al., 2018).  The diurnal variation of the TVOCs, alkenes, aromatics, halocarbons, OVOCs, and alkanes concentrations showed a double-hump structure (Fig. 2a, b, d, & e). This double-hump pattern indicates the contribution of traffic emission during the rush-hours in the morning and evening. The lowest concentration of the TVOCs and different VOC-groups reached 12:00-16:00. Oppositely, the highest concentration of $O_3$ reached at that period (Fig. 3). The lowest $O_3$ concentrations were observed in winter which was consistent with the solar radiations.

**3.3 Sources of VOCs**

~~The use of the toluene/benzene (T/B) ratio is one of the simplest ways to preliminary analyse the VOC sources. If the T/B ratio is < 2, the study area is mainly affected by vehicle emissions (Hui et al., 2018, 2019). If the T/B ratio is > 2, the study area is influenced by other sources (e.g. industry, solvent utilization) beside vehicle emissions (Kumar et al., 2018; Niu et al., 2012; Li et al., 2019). Moreover, the T/B ratios are ranged between 0.2-0.6 in coal and biomass burning affected areas (Wang et al., 2009; Akagi et al., 2011). The diurnal variations in T/B ratios during different seasons are depicted in Fig. 4 (a, b, c, & d). The mean values of T/B ratios were ranged between 0.9-2 (1.4±0.3), 1.3-2 (1.7±0.2), 1.1-1.6 (1.4±0.1), and 1.4-2.7 (1.9±0.3) during summer, autumn, winter, and spring, respectively. As the mean values of T/B ratios were around~~

2, the study area could be mainly affected by vehicle emissions. The double-hump pattern in the diurnal variations in T/B ratios also indicates that the rush-hour traffic had a significant influence on the VOCs concentrations in the study area. Besides, the 75[th] percentiles of T/B ratios were above 2 most of the investigation periods, therefore, the study area could also be influenced by industrial emissions.

Figure 4 (e, f, g, & h) shows the ratios of different alkanes and aromatics to acetylene. Acetylene is a tracer of combustion sources, the ratios of different alkanes and aromatics to acetylene are used to comprehend the contribution of other sources to combustion sources. The mean ratios of propane, n-butane, and i-butane to acetylene were around 2.0-4.0, 0.7-1.6, and 0.4-0.8, respectively during all the seasons, which were smaller than those (11.5, 1.8, and 2.6, respectively) observed in Guangzhou city centre, which was affected by liquefied petroleum gas (LPG) emissions (Zhang et al., 2013). Therefore, LPG usages probably contributed a little fraction to the alkanes in the study area. The mean ratios of benzene, toluene, C8-aromatics, and C9-aromatics to acetylene were around 0.3-1.0, 0.4-1.1, 0.2-0.6, and 0.1, respectively during all the seasons. The observed ratios of benzene and toluene to acetylene were much higher than those found in Jianfeng Mountains in Hainan (0.2 and 0.1, respectively) but comparable to those measured in urban Guangzhou (0.4 and 0.4-1, respectively) (Tang et al., 2007). Besides, the observed ratios of C8-aromatics and C9-aromatics to acetylene were comparable to traffic emission influenced urban Guangzhou (0.68 and 0.2, respectively) and Wuhan (0.5 and 0.2, respectively) (Zhang et al., 2013; Hui et al., 2018). Therefore, vehicle exhaust probably contributed significantly to the aromatics in the study area.

**3.3.2 Potential Source Contribution Function (PSCF)**

Besides the local sources, both the long and short distance transport of air mass could bring VOCs to the study area. Figure 5 shows the wind cluster and PSCF analysis results for different seasons. During summer, the major air masses were was short-distance transports from the southwest (4044%) direction and two long distance transports from southeast (3931 and 25%) 
[revised manuscript text omitted]

[Figure]

(a) Summer

Solvent usage 11.1%
Vehicle emission 3 11.6%
Vehicle emission 2 11.9%
Gasoline evaporation 16.6%
Industrial process and combustion 23.5%
Biogenic source 6.8%
Industrial source 1 8%
Vehicle emission 1 10.6%

(b) Autumn

Vehicle emission 2 11.6%
Gasoline evaporation 9.8%
Biomass burning 7.3%
Industrial process and combustion 32.2%
Industrial source 1 5.3%
Industrial source 2 9%
Vehicle emission 1 14.5%
Solvent usage 10.1%

(c) Winter

Vehicle emission 2 8.1%
Industrial source 2 13.1%
LPG/NG usage 11.2%
Industrial process and combustion 22.8%
Multiple sources 10.1%
Industrial source 1 8.8%
Solvent usage 10.3%
Vehicle emission 1 15.6%

(d) Spring

Industrial source 1 14.2%
LPG/NG usage 18.2%
Industrial source 2 9%
Multiple sources 21.3%
Vehicle emission 2 10%
Gasoline evaporation 5.7%
Vehicle emission 1 17.3%
Solvent usage 4.3%

[revised manuscript text omitted]

**S1. Source apportionment of VOCs**

Figure S1 shows the source profile of summertime VOCs obtained from the PMF model. The resolved factors were identified as biomass/biofuel burning, LPG/NG usage, gasoline evaporation, gasoline vehicle exhaust, diesel vehicle exhaust, industrial production, paint solvent usage, and biogenic source. Factor 1 was characterized by high concentrations of ethane and ethylene. These compounds are tracers of incomplete combustion which emitted from vehicle exhaust and biomass/biofuel burning (An et al., 2017). Benzene, toluene, pentane, and decane concentrations were low in factor 1, therefore, it was identified as biomass/biofuel burning. Factor 2 was distinguished by a significant presence of LPG/NG VOCs propane, isobutene, and n-butane (Shao et al., 2016). So, factor 2 was identified as LPG/NG usage. Factor 3 was dominated by high concentrations of isopentane, n-pentane, and MTBE. Therefore, factor 3 was identified as gasoline evaporation (Song et al., 2018; Wang et al., 2016). Factor 4 possessed high concentrations of vehicle exhaust VOCs benzene and toluene (Song et al., 2018). Although these VOCs are also emitted by industrial processes, the contribution of benzene was several folds higher than toluene in this factor. Therefore, factor 4 was related to vehicle exhaust emission and it was assigned to gasoline vehicle exhaust (An et al., 2017). Factor 5 was characterized by high concentrations of acetylene, n-heptane, and decane. These are related to vehicle emission, especially diesel vehicle exhaust emission as diesel engine produce more acetylene than the gasoline engine does (Song et al., 2018; An et al., 2017). Therefore, factor 5 was attributed to diesel vehicle exhaust. Factor 6 was dominated by toluene and the sampling site was beside an industrial area. So, we identified this factor as industrial production. Due to the high contribution of o-xylene, m,p-xylene, ethylbenzene and styrene, factor 7 was assigned to paint solvent usage sources (Li et al., 2018). Factor 8 was attributed to the biogenic source, which was mainly distinguished by a high concentration of isoprene (Song et al., 2018).

During autumn, the possible VOC sources were biomass/biofuel burning, multiple sources, gasoline vehicle exhaust, vehicle emission, LPG/NG usage, paint solvent usage and gasoline evaporation (Fig.S2). Factor 1 was represented by a high concentration of ethane and ethylene, so, it was identified as biomass/biofuel burning (An et al., 2017). Factor 2 was dominated by isoprene, n-heptane, decane, and acetylene. Among these compounds, isoprene is mainly emitted by trees and the rest of the compounds are related to diesel vehicle exhaust emission. Therefore,

Factor 2 was identified as multiple sources. Factor 3 was identified as gasoline vehicle exhaust due to the high contribution of benzene and a relatively small contribution of toluene. In factor 3, benzene was several folds higher than toluene. The 4th factor was mainly composed of vehicle emission-related compounds 2-methyl pentane, n-hexane, n-heptane, n-pentane, and isopentane, therefore, identified as vehicle emission (Song et al., 2018). Factor 5 was assigned to LPG/NG usage as propane, isobutene, and n-butane were the main contributor to it (Shao et al., 2016). Factor 6 was characterized by a high concentration of o-xylene, m,p-xylene, ethylbenzene and styrene, which are typical tracer of paint solvent usage. Factor 7 was identified as gasoline evaporation, it was dominated by high concentrations of isopentane, n-pentane, and MTBE (Song et al., 2018).

During winter, the source factors were identified as gasoline vehicle exhaust, vehicle exhaust, gasoline evaporation, biomass/biofuel burning, multiple sources, LPG/NG usage, and paint solvent usage (Fig. S3). Factor 1 was assigned to gasoline vehicle exhaust; it was dominated by benzene and toluene and the contribution of benzene was twice of toluene. Factor 2 was represented by a high concentration of isobutene, n-butane, acetylene, ethylene, ethane, n-heptane and decane. Isobutene and n-butane are related to LPG/NG usage, but, the contribution of propane was zero in factor 2. Acetylene, ethylene, and ethane emitted from combustion sources like vehicle exhaust and biomass burning. Decane and n-heptane are related to vehicle emission. By considering the above information, factor 2 was identified as vehicle exhaust. Factor 3 was identified as gasoline evaporation and it was characterized by high concentrations of isopentane and n-pentane. Factor 4 was characterized by a high contribution of ethylene and ethane; therefore, it was identified as biomass/biofuel burning. Factor 5 was characterized by high concentrations of isoprene, propane, n-hexane and n-heptane. Propane is related to LPG/NG usage, isoprene mainly emitted from trees (evergreen trees in winter), and n-hexane and n-heptane are related to vehicle emission. By considering the above information, factor 5 was assigned to multiple sources. Factor 6 was dominated by high concentrations of propane. Therefore, it was identified as LPG/NG usage. Factor 7 was identified as paint solvent usage due to the high contribution of o-xylene, m,p-xylene, ethylbenzene and styrene (Zhang et al., 2018; Song et al., 2020).

During spring, the possible VOC sources were biomass/biofuel burning, paint solvent usage, multiple sources, gasoline evaporation, gasoline vehicle exhaust, LPG/NG usage, and diesel vehicle exhaust (Fig. S4). Factors 1 was identified as a biomass/biofuel burning source for the high loading of ethylene and ethane and relatively lower contribution from the vehicle emission related compounds. Due to the high contribution of o-xylene, styrene, m,p-xylene, and ethylbenzene, factor 2 was assigned to paint solvent usage sources (Li et al., 2018). Factor 3 had a high contribution of isoprene, n-hexane, n-heptane, decane, MTBE, toluene, ethylbenzene, and o-xylene. Therefore, factor 3 was identified as multiple sources. Factor 4 was represented by a high concentration of isopentane, n-pentane, and MTBE. Therefore, factor 4 was identified as gasoline evaporation. Factor 5 was represented by high concentrations of benzene, therefore, identified as gasoline vehicle exhaust. 
[revised manuscript text omitted]

**Authors' response to referee 3**

The authors are grateful to the anonymous referee #3 for his/her constructive comments and suggestions. Here, we present the response for each of the comments. The revised manuscript and supplement with tracked changes can be found at the end of the document.

1. **PMF analysis-**

● **PMF was performed separately on each season. Was there enough data in during 2020 to draw meaningful conclusions? It is difficult to tell the duration of measurements from figure 1.**

Authors' response: We presented 20 days data (464 data points) for the spring season in 2020 (April 15 - May 4). Due to the instruments failure, we could not continue the measurements for a longer period. Still, PMF model analysis on 464 hourly data points is enough to draw a meaningful conclusion.

● **Do the PMF results indicate that there is seasonality in source sector emissions, or are the fingerprints consistent?**

Authors' response: Yes, the PMF results indicate that there is seasonality in source sector emissions, for instance, (i) contribution from gasoline evaporation was much higher in summer than those in autumn and spring (ii) biogenic source was clearly distinguished in summer (due to high emission), but mixed with other sources during the rest of the seasons.

● **Why was an 8 factor solution chosen (v. 7 v. 6 factors, etc)?**

Authors' response: The reason for choosing 7 to 8-factor solution has been mentioned in the material and methods (section 2.2) of the revised manuscript (added at the end of this document).

● **The supplemental figures are difficult to read due to the number of species included in the figure. It should be noted which axis should be read with each dataset.**

Authors' response: Both the issues have been solved in the revised manuscript.

● **As noted by reviewer 2, the results are odd (e.g., biogenic isopentane; propane from**

**solvents). The revised and trimmed-down PMF submitted in response to the reviewer still has these oddities.**

Authors' response: It has been solved in the revised manuscript added below. For instance, isopentane is assigned to gasoline evaporation and propane is assigned to LPG/NG usages sources.

**2. PSCF analysis-**

**● I do not think this method is ubiquitous enough to merit the brevity of explanation. For example, what do the colors mean in figure 5? What does the impact of a 24 v. 48 v. 72 h back-trajectory have on the analysis? Does this assume that VOCs have the same lifetime as the back-trajectory?**

Authors' response: PSCF has been widely used to determine potential source areas of particulate pollutants, and in recent years, it has been also used to determine the potential source areas of VOCs (Hao et al., 2019; Ming et al., 2017; Sun et al., 2015; Song et al., 2018; Hui et al., 2019). The PSCF value itself is a probability (from 0 to 1) indicating the probability of being a possible source of a pollutant (e.g. colorbar in Figure 5). This probability is calculated by dividing the number of trajectories with a high air pollutant concentration (higher than the mean TVOCs concentration in our case) by the total number of trajectories at the receptor. Areas in Figure 5 with higher PSCF values indicate that the corresponding region is a potential source region of severe VOC pollution. We generally choose backward trajectories of different duration depending on the studied species, for example, if it is an inert pollutant such as Black Carbon, a backward trajectory of 120h or longer will be chosen. The biggest difference between 24h vs 48h vs 72h analysis is that we assume different atmospheric lifetimes of the pollutants, so that the potential source area obtained is also different, e.g., the range of 72h is larger and the high probability range (high PSCF value) is larger too, but if the pollutants themselves are not able to remain for 72h, there will be some bias in the estimation of the potential source area.

**● It is not clear what useful/new information is derived from this analysis. The main results of this analysis (line 428-430) seems to be that the VOCs measured at the site**

come primarily come from industries located near the site. Would there be a reason to think otherwise?

Authors' response: Long-distance transport of VOC pollution to the receptor site is not uncommon (Derstroff et al., 2017; Patokoski et al., 2015). So, it is not always the case that the VOC should come to the measurement site from the nearby areas.

**3. OFP/EKMA/RIR**
● **Discussion of wintertime OFP seems unusual, as photochemical ozone production in the winter is not a primary concern.**

Authors' response: The OFP analysis has already been removed following the suggestion of referee #2.

● **There are not enough details in the description of the F0AM model. For example: Which VOCs were constrained (not all are in MCM)?**

Authors' response: More information has been added in the methods, and the list of constrained VOCs has been added in Table S1.

**Other comments-**

● **I find the title "High Contributions of Halohydrocarbon and Aromatic Compounds to Emissions and Chemistry of Atmospheric VOCs in Industrial Area" to be misleading. Halocarbons do not contribute significantly to either SOA or ozone chemistry. Neither halocarbons nor aromatics dominate calculated OFP in the summer (Figure 5).**

Authors' response: The title has been modified in the revised version.

● **It seems a major conclusion (line 431) is that industries and vehicles should have the most priority in reducing emissions. Nearly all anthropogenic VOCs in this area could**

be classified as industry/vehicles, so this conclusion seems too vague to be meaningful. Is it a new finding?

Authors' response: This part has been modified in the revised manuscript. According to the revised PMF model analysis vehicle-related emissions were the major VOC sources in the industry affected area.

• **It is unusual to see the notation KOH (typically lowercase/supscripts are for kinetics, uppercase for equilibrium reactions).**

Authors' response: Amended.

• **There are no units in supplemental tables.**

Authors' response:  Units have been added in the supplemental tables.

• **It is not clear how to interpret numbers given in lines 272 (e.g., 0.9-2 (1.4 +/- 0.3))**

Authors' response:  It was the range and mean value of toluene to benzene ratio in ppbv/ppbv unit. The toluene to benzene ratio analysis section has already been removed following the suggestion of referee #2.

[revised manuscript text omitted]

(a) Summer

- Solvent usage 11.1%
- Vehicle emission 3 11.6%
- Vehicle emission 2 11.9%
- Gasoline evaporation 16.6%
- Industrial process and combustion 23.5%
- Biogenic source 6.8%
- Industrial source 1 8%
- Vehicle emission 1 10.6%

(b) Autumn

- Vehicle emission 2 11.6%
- Gasoline evaporation 9.8%
- Biomass burning 7.3%
- Industrial process and combustion 32.2%
- Industrial source 1 5.3%
- Industrial source 2 9%
- Vehicle emission 1 14.5%
- Solvent usage 10.1%

(c) Winter

- Vehicle emission 2 8.1%
- Industrial source 2 13.1%
- Industrial process and combustion 22.8%
- LPG/NG usage 11.2%
- Multiple sources 10.1%
- Industrial source 1 8.8%
- Solvent usage 10.3%
- Vehicle emission 1 15.6%

(d) Spring

- Industrial source 1 14.2%
- LPG/NG usage 18.2%
- Industrial source 2 9%
- Multiple sources 21.3%
- Vehicle emission 2 10%
- Gasoline evaporation 5.7%
- Vehicle emission 1 17.3%
- Solvent usage 4.3%

[Figure]

930    **Figure 6: relative contributions of different sources to ambient VOCs in Nanjing industrial area during different seasons**

[Figure]

Figure 7: Contribution to OH loss rates of different VOC-groups and the top 10 VOC species in different seasons

[Figure]

**Figure 8: Contribution to ozone formation potential of different VOC-groups and the top 10 VOC species in different seasons**

[Figure]

**Figure 9: Contribution to secondary organic aerosol formation potential of different VOC-groups and the top 10 VOC species in different seasons**

[Figure]

**Figure 10: O₃ isopleth diagram for (a) summer (b) autumn, (c) winter, and (d) spring based on percentage changes in VOCs and NOx concentrations in Nanjing and corresponding modelled O₃ production.**

950

[Figure]

**Figure 11: The RIR values of the VOC, NOx, and CO for the different seasons in Nanjing**

955

960

965

**Measurement report: High Contributions of Halocarbon and Aromatic Compounds to Atmospheric VOCs in Industrial Area**

Ahsan Mozaffar [1,2,3], Yan-Lin Zhang [1,2,3*], Yu-Chi Lin[1,2,3], Feng Xie [1,2,3], Mei-Yi Fan [1,2,3], and

Fang Cao [1,2,3]

[1]Yale-NUIST Center on Atmospheric Environment, International Joint Laboratory on Climate and Environment Change, Nanjing University of Information Science and Technology, Nanjing, 210044, China.

[2]Key Laboratory Meteorological Disaster; Ministry of Education & Collaborative Innovation Center on Forecast and Evaluation of Meteorological Disaster, Nanjing University of Information Science and Technology, Nanjing, 210044, China.

[3]Jiangsu Provincial Key Laboratory of Agricultural Meteorology, College of Applied Meteorology, Nanjing University of Information Science & Technology, Nanjing 210044, China.

*Correspondence to*: Yan-Lin Zhang (dryanlinzhang@outlook.com)

**Table S1: OH reaction rate constant ($K_{OH}$) of VOCs and list of VOCs constrained in the F0AM model**

| Compounds | $K_{OH}$ (cm$^3$ molecule$^{-1}$ s$^{-1}$) (Carter, 2010) | Constrained in F0AM model |
|---|---|---|
| Ethane | 2.54E-13 | Yes |
| propane | 1.11E-12 | Yes |
| isobutane | 2.14E-12 | Yes |
| n-butane | 2.38E-12 | Yes |
| isopentane | 3.60E-12 | Yes |
| n-pentane | 3.84E-12 | Yes |
| 2,2 dimethylbutane | 2.27E-12 | |
| 2,3 dimethyl butane | 5.79E-12 | |
| 2-methyl pentane | 5.20E-12 | Yes |
| cyclopentane | 5.02E-12 | |

| | | |
|---|---|---|
| 3-methylpentane | 5.20E-12 | |
| n-hexane | 5.25E-12 | Yes |
| 2,4-dimethylpentane | 4.77E-12 | |
| methylcyclopentane | 5.68E-12 | |
| isoheptane | 6.81E-12 | Yes |
| cyclohexane | 7.02E-12 | |
| 2,3-dimethylpentane | 7.15E-12 | |
| 3-methylhexane | 7.17E-12 | Yes |
| 2,2,4-trimethylpentane | 3.38E-12 | |
| heptane | 6.81E-12 | Yes |
| methylcyclohexane | 9.64E-12 | |
| 2-methylheptane | 8.31E-12 | |
| n-octane | 8.16E-12 | Yes |
| n-nonane | 9.75E-12 | Yes |
| Decane | 1.10E-11 | Yes |
| n-hendecane | 1.23E-11 | Yes |
| dodecane | 1.32E-11 | Yes |
| ethylene | 8.15E-12 | Yes |
| Propylene | 2.60E-11 | Yes |
| trans-2-butene | 6.32E-11 | Yes |
| cis-2-butene | 5.58E-11 | |
| 1-butene | 3.11E-11 | Yes |
| 1,3- butadeine | 6.59E-11 | |
| 1-pentene | 3.14E-11 | Yes |
| tran-2-pentene | 6.70E-11 | |
| isoprene | 9.96E-11 | Yes |
| cis-2-pentene | 6.50E-11 | Yes |
| 1-hexene | 3.70E-11 | Yes |
| acetylene | 7.56E-13 | Yes |
| benzene | 1.22E-12 | Yes |
| toluene | 5.58E-12 | Yes |

| | | |
|---|---|---|
| ethylbenzene | 7.00E-12 | Yes |
| m,p-xylene | 2.31E-11 | Yes |
| o-xylene | 1.36E-11 | Yes |
| Styrene | 5.80E-11 | Yes |
| Cumene | 6.30E-12 | Yes |
| n-propylbenzene | 5.80E-12 | Yes |
| 3-ethyltoulene | 1.86E-11 | Yes |
| 4-ethyltoulene | 1.18E-11 | Yes |
| Mesitylene | 5.67E-11 | Yes |
| 2-ethyltoulene | 1.19E-11 | Yes |
| 1,2,4-trimethylbenzene | 3.25E-11 | Yes |
| 1,2,3-trimethylbenzene | 3.27E-11 | Yes |
| 1,3-diethylbenzene | 2.55E-11 | |
| 1,4-diethylbenzene | 1.64E-11 | Yes |
| Naphthalene | 2.30E-11 | |
| Chloromethane | 4.48E-14 | Yes |
| vinyl chloride | 6.90E-12 | Yes |
| methyl bromide | 4.12E-14 | |
| Chloroethene | 0 | Yes |
| trichlorofloromethane | 0 | |
| Vinylidene chloride | 0 | |
| 1,1,2-Trichlor-1,2,2-trifluorethan | 0 | |
| Dichloromethane | 1.45E-13 | Yes |
| trans-1,2-dichloroethylene | 0 | |
| 1,1-dichloroethane | 2.60E-13 | Yes |
| cis-1,2-dichloroethylene | 0 | |
| Chloroform | 1.06E-13 | Yes |
| carbon tetrachloride | 0 | |
| 1,2-dichloroethane | 2.53E-13 | Yes |

| | | |
|---|---|---|
| Trichloroethylene | 2.34E-12 | Yes |
| 1,2-dichloropropane | 4.50E-13 | Yes |
| bromodichloromethane | 0 | |
| trans-1,3-dichloropropene | 1.44E-11 | |
| cis-1,3-dichloropropene | 8.45E-12 | |
| 1,1,2-trichloroethane | 2.00E-13 | Yes |
| tetrachloroethylene | 0 | Yes |
| 1,2-dibromoethane | 2.27E-13 | Yes |
| Chlorobenzene | 7.70E-13 | |
| Bromoform | 0 | |
| 1,1,2,2-tetrachloroethane | 0 | Yes |
| 1,3-dichlorobenzene | 5.55E-13 | |
| 1,4 dichlorobebezne | 5.55E-13 | |
| benzyl chloride | 0 | |
| 1,2-dichlorobenzene | 5.55E-13 | |
| 1,2,4-trichlorobenzene | 0 | |
| hexachloro-1,3-butadiene | 0 | |
| carbon disulfide | 2.76E-12 | |
| Acrolein | 1.99E-11 | Yes |
| Acetone | 1.91E-13 | Yes |
| Isopropanol | 5.09E-12 | Yes |
| MTBE | 0 | Yes |
| vinyl acetate | 3.16E-11 | |
| MEK | 1.20E-12 | Yes |
| ethyl acetate | 1.60E-12 | Yes |
| Tetrahydrofuran | 1.61E-11 | |
| methyl methacrylate | 5.25E-11 | |
| 1,4-dioxane | 3.83E-11 | |
| 4-methyl-2-pentanone | 1.27E-11 | Yes |
| 2-hexanone | 9.10E-12 | Yes |

**Table S2: VOC concentrations (ppbv) measured in the industrial area in Nanjing. VOC concentrations observed in**

**previous studies in Nanjing are also listed.**

[revised manuscript text omitted]

**S1. Source apportionment of VOCs**

Figure S1 shows the source profile of summertime VOCs obtained from the PMF model. The resolved factors were identified as biomass/biofuel burning, LPG/NG usage, gasoline evaporation, gasoline vehicle exhaust, diesel vehicle exhaust, industrial production, paint solvent usage, and biogenic source. Factor 1 was characterized by high concentrations of ethane and ethylene. These compounds are tracers of incomplete combustion which emitted from vehicle exhaust and biomass/biofuel burning (An et al., 2017). Benzene, toluene, pentane, and decane concentrations were low in factor 1, therefore, it was identified as biomass/biofuel burning. Factor 2 was distinguished by a significant presence of LPG/NG VOCs propane, isobutene, and n-butane (Shao et al., 2016). So, factor 2 was identified as LPG/NG usage. Factor 3 was dominated by high concentrations of isopentane, n-pentane, and MTBE. Therefore, factor 3 was identified as gasoline evaporation (Song et al., 2018; Wang et al., 2016).  Factor 4 possessed high concentrations of vehicle exhaust VOCs benzene and toluene (Song et al., 2018). Although these VOCs are also emitted by industrial processes, the contribution of benzene was several folds higher than toluene in this factor. Therefore, factor 4 was related to vehicle exhaust emission and it was assigned to gasoline vehicle exhaust (An et al., 2017). Factor 5 was characterized by high concentrations of acetylene, n-heptane, and decane. These are related to vehicle emission, especially diesel vehicle exhaust emission as diesel engine produce more acetylene than the gasoline engine does (Song et al., 2018; An et al., 2017). Therefore, factor 5 was attributed to diesel vehicle exhaust. Factor 6 was dominated by toluene and the sampling site was beside an industrial area. So, we identified this factor as industrial production. Due to the high contribution of o-xylene, m,p-xylene, ethylbenzene and styrene, factor 7 was assigned to paint solvent usage sources (Li et al., 2018). Factor 8 was attributed to the biogenic source, which was mainly distinguished by a high concentration of isoprene (Song et al., 2018).

During autumn, the possible VOC sources were biomass/biofuel burning, multiple sources, gasoline vehicle exhaust, vehicle emission, LPG/NG usage, paint solvent usage and gasoline evaporation (Fig.S2). Factor 1 was represented by a high concentration of ethane and ethylene, so, it was identified as biomass/biofuel burning (An et al., 2017). Factor 2 was dominated by isoprene, n-heptane, decane, and acetylene. Among these compounds isoprene is mainly emitted by trees and rest of the compounds are related to diesel vehicle exhaust emission. Therefore, Factor 2 was identified as multiple sources. Factor 3 was identified as gasoline vehicle exhaust

due to high contribution of benzene and toluene. In factor 3, benzene was several folds higher than toluene. The 4[th] factor was mainly composed of vehicle emission related compounds 2-methyl pentane, n-hexane, n-heptane, n-pentane, and isopentane, therefore, identified as vehicle emission (Song et al., 2018). Factor 5 was assigned to LPG/NG usage as propane, isobutene, and n-butane were the main contributor to it (Shao et al., 2016). Factor 6 was characterized by a high concentration of o-xylene, m,p-xylene, ethylbenzene and styrene, which are typical tracer of paint solvent usage. Factor 7 was identified as gasoline evaporation, it was dominated by high concentrations of isopentane, n-pentane, and MTBE (Song et al., 2018).

During winter, the source factors were identified as gasoline vehicle exhaust, vehicle exhaust, gasoline evaporation, biomass/biofuel burning, multiple sources, LPG/NG usage, and paint solvent usage (Fig. S3). Factor 1 was assigned to gasoline vehicle exhaust; it was dominated by benzene and toluene and contribution of benzene was twice of toluene. Factor 2 was represented by a high concentration of isobutene, n-butane, acetylene, ethylene, ethane, n-heptane and decane. Isobutene and n-butane are related to LPG/NG usage, but, the contribution of propane was zero in factor 2. Acetylene, ethylene, and ethane emitted from combustion sources like vehicle exhaust and biomass burning. Decane and n-heptane are related to vehicle emission. By considering above information, factor 2 was identified as vehicle exhaust. Factor 3 was identified as gasoline evaporation and it was characterized by high concentrations of isopentane and n-pentane. Factor 4 was characterized by high contribution of ethylene and ethane; therefore, it was identified as biomass/biofuel burning. Factor 5 was characterized by high concentrations of isoprene, propane, n-hexane and n-heptane. Propane is related to LPG/NG usage, isoprene mainly emitted from trees (evergreen trees in winter), and n-hexane and n-heptane are related to vehicle emission. By considering above information, factor 5 was assigned to multiple sources. Factor 6 was dominated by high concentrations of propane. Therefore, it was identified as LPG/NG usage. Factor 7 was identified as paint solvent usage due to the high contribution of o-xylene, m,p-xylene, ethylbenzene and styrene (Zhang et al., 2018; Song et al., 2020).

During spring, the possible VOC sources were biomass/biofuel burning, paint solvent usage, multiple sources, gasoline evaporation, gasoline vehicle exhaust, LPG/NG usage, and diesel vehicle exhaust (Fig. S4). Factors 1 was identified as a biomass/biofuel burning source for the high loading of ethylene and ethane and relatively lower contribution from the vehicle emission related compounds. Due to the high contribution of o-xylene, styrene, m,p-xylene, and

ethylbenzene, factor 2 was assigned to paint solvent usage sources (Li et al., 2018). Factor 3 had a high contribution of isoprene, n-hexane, n-heptane, decane, MTBE, toluene, ethylbenzene, and o-xylene. Therefore, factor 3 was identified as multiple sources. Factor 4 was represented by a high concentration of isopentane, n-pentane, and MTBE. Therefore, factor 4 was identified as gasoline evaporation. Factor 5 was represented by high concentrations of benzene, therefore, identified as gasoline vehicle exhaust. 
[revised manuscript text omitted]

1170

**Authors' response to referee 4**

The authors are grateful to anonymous referee #4 for the insightful comments and suggestions. Here, we present the answers for each of the comments. The revised manuscript and supplement with tracked changes can be found at the end of the document.

**Approximately 100 speciated VOCs, measured by GC/FID/MS, are reported from a Nanjing industrial area in China from July 2018 – May 2020. The non-continuous measurement periods include field data from summer, autumn, winter, and spring. This measurement report focuses on the inclusion of select halocarbons and oxygenated VOCs to a "total" VOC (TVOC) measurement, which is then compared to past TVOC studies in Nanjing and other Chinese cities. The authors performed data analysis techniques including PMF, PSCF, and photochemical box modeling on this data set to assess potential VOC sources and the impact of these VOCs on local ozone production.**

**General Comments**

**I note that the authors have already heavily revised their manuscript in response to previous Referee Comments (e.g. PMF results and discussion, exclusion of VOC ratio and ozone formation potential discussions) and this review is primarily based on the revised version.**
**The observations reported here, speciated and quantified VOC composition in Nanjing, wuold be a useful resource for the atmospheric research community. While the measurement techniques (described in Mozaffar et al. 2020, Atmospheric Research) employed for this study are appropriate and appear well done, some of the analysis and interpretations included in this measurement report could be refined. The following are some areas for concern that they authors might consider revising:**

- **Throughout the manuscript there is continuous discussion and comparison of VOC compositions measured in various cities from different studies. Here the authors compare observations of certain classes of VOCs by describing them as "%" (e.g. lines 75 – 91). I find this to not only be confusing, but also not a useful metric, as each study did not measure the same suite of VOCs. For example, comparing this studies % contribution of alkanes to the total VOC measurement to another studies is not useful if the other study was not measuring the same total VOC list. I would recommend that the authors revise the manuscript throughout. If they would like to directly compare their measurements to previous reports, they should do so on a concentration basis. Until this comparision is revised it is difficult to gauge how the measurements reported here compare to other areas.**

Authors' response: VOC concentrations are reported now instead of "%".

- **Along with the above comment, the use of the term "TVOCs" throughout this manuscript I find to be troubling. The TVOC measurement here is a sum of the suite of**

**VOCs measured, but it is not a total VOC measurement. This term is especially frustrating in section 3.2, lines 242 – 263, where the authors compare their TVOC with that from other cities where entire groups of VOCs (e.g. halocarbons) were not included. The manuscript should be revised with care to make sure that any quantitative comparison with previous studies is "apples to apples" (even if that means that not all of the VOCs reported in this study are included in a specific comparison in the discussion).**

Authors' response: In the abstract, the term "TVOCs" is now defined. Hopefully, it will eliminate the confusion. In this kind of study, the term "TVOCs" is widely used to present the sum of the suite of VOCs measured.

The VOC species we presented in our manuscript does not exactly match with the previous studies discussed in the comparison section. That is why the "apples to apples" comparison of TVOCs with previous studies is not possible. Therefore, we deleted this TVOCs comparison part in the revised manuscript. However, we modified the TVOCs comparison part for the two previous studies perform in Nanjing. Except for halocarbons and OVOCs, only 4 VOC species concentrations are not presented in these studies compare to the current observation (Table S2). In the revised manuscript, the TVOCs concentration without halocarbons and OVOCs are compared between these studies.

**Specific Comments**

▪ **In section 2.4, which VOCs were used to model the impact of their reduction on ozone? The text (line 175) says 11 VOCs but does not list which ones or the authors' reasoning for those choices.**

Authors' response: 61 VOCs were constrained in the model. These constrained VOCs are listed in the revised Table S1. The rest of the measured VOCs were not constrained in the model as their reactions are not included in MCM. This information is now added in section 2.4. Actually, "11 NOx × 11 VOC concentrations" in line 175 represents something else. The model was run for 11 different concentration scenarios of the constrained VOCs and 11 different concentration scenarios of NOx.

▪ **In section 3.2, lines 224 – 228, the authors attribute a higher contribution of OVOCs in summer/spring to enhanced biogenic emissions. However, the concentration of the reported OVOCs (figure 2c) are fairly constant throughout the year (or even reduced in the summer). It appears the higher contribution of OVOCs in spring/summer is due to the reduction of other classes of VOCs (e.g. halocarbons).**

Authors' response: Many thanks for the comment. These lines have been deleted in the revised manuscript.

▪ **The spring portion of the data set is from April 2020, could the authors provide comment on whether they view this measurement period to be representative of a**

**typical spring in Nanjing or not due to differences in daily operations due to the Covid pandemic.**

Authors' response: Actually, the spring portion of the data set is from April 15 to May 4 2020. The measurement period is representative of a typical spring in Nanjing; the temperature was around 20 °C (Figure 3). For your kind information, we could not do the measurement in spring 2019 due to the COVID pandemic.

- **The conclusion section should be re-written for clarity. The authors have numerous statements that are either redundant or grammatically incorrect.**

Authors' response: The conclusions section has been modified.

**Technical Corrections**

**Line 115: The authors should cite the supplemental from Mozaffar et al., 2020 here to give the reader a resource the GC/FID/MS technique, since the validity of the data included in the report relies on the quality of the analytical measurement**

Authors' response: Done, we cited it in the revised manuscript.

**Line 124: There are two Mozaffar et al., 2020 references in the list. They should be clarified as "a" and "b"**

Authors' response: Actually, there is one "Mozaffar et al., 2020" reference in the list. The other one is "Mozaffar & Zhang, 2020".

**Line 149: I believe the notation for reaction rate constant with OH should be "kOH,i" generally using a capital "K" is for equilibrium constants.**

Authors' response: It is corrected in the revised manuscript and supplement.

**Line 217-218: Which citation are you referring to? There are two Nanjing studies included in Table S2.**

Authors' response: Citations have been added to the text now. Further information about location has also been added in Table S2.

**Figure 2: It would be helpful if the figure spacing was revised so that it is clearer that the TVOC and alkane data are on their own axis.**

Authors' response: Done, a revised figure has been added.

**Figure 3, Figure 4: It could help the comparison if all left and right axes were kept to the same range.**

Authors' response: The axes of Figure 3 have been fixed. Figure 4 has already been removed following the suggestion of Referee #2.

**Revised Figure 6: Keep color scheme consistent between pie graphs**

Authors' response: Done, the previous revised figure has been replaced by a modified one.

110 **Table S2: Missing concentration units (ppb)**

Authors' response: The unit has been added in the caption.

**Throughout: The manuscript should be edited for grammar, spelling errors, and redundant sentences.**

115 Authors' response: The manuscript has been checked thoroughly and edited for grammar, spelling errors, and redundant sentences.

[revised manuscript text omitted]

(a) Summer

Solvent usage 11.1%
Vehicle emission 3 11.6%
Gasoline evaporation 16.6%
Vehicle emission 2 11.9%
Biogenic source 6.8%
Industrial process and combustion 23.5%
Industrial source 1 8%
Vehicle emission 1 10.6%

(b) Autumn

Vehicle emission 2 11.6%
Gasoline evaporation 9.8%
Biomass burning 7.3%
Industrial source 1 5.3%
Industrial process and combustion 32.2%
Industrial source 2 9%
Solvent usage 10.1%
Vehicle emission 1 14.5%

(c) Winter

Vehicle emission 2 8.1%
Industrial source 2 13.1%
Industrial process and combustion 22.8%
LPG/NG usage 11.2%
Multiple sources 10.1%
Industrial source 1 8.8%
Solvent usage 10.3%
Vehicle emission 1 15.6%

(d) Spring

Industrial source 1 14.2%
LPG/NG usage 18.2%
Industrial source 2 9%
Vehicle emission 2 10%
Multiple sources 21.3%
Gasoline evaporation 5.7%
Vehicle emission 1 17.3%
Solvent usage 4.3%

[Figure]

**Figure 6: relative contributions of different sources to ambient VOCs in Nanjing industrial area during different seasons**

915

[Figure]

**Figure 7: Contribution to OH loss rates of different VOC-groups and the top 10 VOC species in different seasons**

920

[Figure]

**Figure 8: Contribution to ozone formation potential of different VOC-groups and the top 10 VOC species in different seasons**

[Figure]

925

**Figure 9: Contribution to secondary organic aerosol formation potential of different VOC-groups and the top 10 VOC species in different seasons**

[Figure]

**Figure 10: O₃ isopleth diagram for (a) summer (b) autumn, (c) winter, and (d) spring based on percentage changes in VOCs and NOx concentrations in Nanjing and corresponding modelled O₃ production.**

[Figure]

**Figure 11: The RIR values of the VOC, NOx, and CO for the different seasons in Nanjing**

**Measurement report: High Contributions of Halocarbon and Aromatic Compounds to Atmospheric VOCs in Industrial Area**

Ahsan Mozaffar [1,2,3], Yan-Lin Zhang [1,2,3*], Yu-Chi Lin[1,2,3], Feng Xie [1,2,3], Mei-Yi Fan [1,2,3], and Fang

Cao [1,2,3]

[1]Yale-NUIST Center on Atmospheric Environment, International Joint Laboratory on Climate and Environment Change, Nanjing University of Information Science and Technology, Nanjing, 210044, China.

[2]Key Laboratory Meteorological Disaster; Ministry of Education & Collaborative Innovation Center on Forecast and Evaluation of Meteorological Disaster, Nanjing University of Information Science and Technology, Nanjing, 210044, China.

[3]Jiangsu Provincial Key Laboratory of Agricultural Meteorology, College of Applied Meteorology, Nanjing University of Information Science & Technology, Nanjing 210044, China.

*Correspondence to*: Yan-Lin Zhang (dryanlinzhang@outlook.com)

**Table S1: OH reaction rate constant ($k_{OH}$) of VOCs and list of VOCs constrained in the F0AM model**

| Compounds | $k_{OH}$ (cm$^3$ molecule$^{-1}$ s$^{-1}$) (Carter, 2010) | Constrained in F0AM model |
|---|---|---|
| Ethane | 2.54E-13 | Yes |
| propane | 1.11E-12 | Yes |
| isobutane | 2.14E-12 | Yes |
| n-butane | 2.38E-12 | Yes |
| isopentane | 3.60E-12 | Yes |
| n-pentane | 3.84E-12 | Yes |
| 2,2 dimethylbutane | 2.27E-12 | |
| 2,3 dimethyl butane | 5.79E-12 | |
| 2-methyl pentane | 5.20E-12 | Yes |
| cyclopentane | 5.02E-12 | |
| 3-methylpentane | 5.20E-12 | |
| n-hexane | 5.25E-12 | Yes |
| 2,4-dimethylpentane | 4.77E-12 | |
| methylcyclopentane | 5.68E-12 | |
| isoheptane | 6.81E-12 | Yes |

| | | |
|---|---|---|
| cyclohexane | 7.02E-12 | |
| 2,3-dimethylpentane | 7.15E-12 | |
| 3-methylhexane | 7.17E-12 | Yes |
| 2,2,4-trimethylpentane | 3.38E-12 | |
| heptane | 6.81E-12 | Yes |
| methylcyclohexane | 9.64E-12 | |
| 2-methylheptane | 8.31E-12 | |
| n-octane | 8.16E-12 | Yes |
| n-nonane | 9.75E-12 | Yes |
| Decane | 1.10E-11 | Yes |
| n-hendecane | 1.23E-11 | Yes |
| dodecane | 1.32E-11 | Yes |
| ethylene | 8.15E-12 | Yes |
| Propylene | 2.60E-11 | Yes |
| trans-2-butene | 6.32E-11 | Yes |
| cis-2-butene | 5.58E-11 | |
| 1-butene | 3.11E-11 | Yes |
| 1,3- butadeine | 6.59E-11 | |
| 1-pentene | 3.14E-11 | Yes |
| tran-2-pentene | 6.70E-11 | |
| isoprene | 9.96E-11 | Yes |
| cis-2-pentene | 6.50E-11 | Yes |
| 1-hexene | 3.70E-11 | Yes |
| acetylene | 7.56E-13 | Yes |
| benzene | 1.22E-12 | Yes |
| toluene | 5.58E-12 | Yes |
| ethylbenzene | 7.00E-12 | Yes |
| m,p-xylene | 2.31E-11 | Yes |
| o-xylene | 1.36E-11 | Yes |
| Styrene | 5.80E-11 | Yes |
| Cumene | 6.30E-12 | Yes |
| n-propylbenzene | 5.80E-12 | Yes |
| 3-ethyltoulene | 1.86E-11 | Yes |
| 4-ethyltoulene | 1.18E-11 | Yes |
| Mesitylene | 5.67E-11 | Yes |
| 2-ethyltoulene | 1.19E-11 | Yes |
| 1,2,4-trimethylbenzene | 3.25E-11 | Yes |
| 1,2,3-trimethylbenzene | 3.27E-11 | Yes |
| 1,3-diethylbenzene | 2.55E-11 | |
| 1,4-diethylbenzene | 1.64E-11 | Yes |
| Naphthalene | 2.30E-11 | |

| | | |
|---|---|---|
| Chloromethane | 4.48E-14 | Yes |
| vinyl chloride | 6.90E-12 | Yes |
| methyl bromide | 4.12E-14 | |
| Chloroethene | 0 | Yes |
| trichlorofloromethane | 0 | |
| Vinylidene chloride | 0 | |
| 1,1,2-Trichlor-1,2,2-trifluorethan | 0 | |
| Dichloromethane | 1.45E-13 | Yes |
| trans-1,2-dichloroethylene | 0 | |
| 1,1-dichloroethane | 2.60E-13 | Yes |
| cis-1,2-dichloroethylene | 0 | |
| Chloroform | 1.06E-13 | Yes |
| carbon tetrachloride | 0 | |
| 1,2-dichloroethane | 2.53E-13 | Yes |
| Trichloroethylene | 2.34E-12 | Yes |
| 1,2-dichloropropane | 4.50E-13 | Yes |
| bromodichloromethane | 0 | |
| trans-1,3-dichloropropene | 1.44E-11 | |
| cis-1,3-dichloropropene | 8.45E-12 | |
| 1,1,2-trichloroethane | 2.00E-13 | Yes |
| tetrachloroethylene | 0 | Yes |
| 1,2-dibromoethane | 2.27E-13 | Yes |
| Chlorobenzene | 7.70E-13 | |
| Bromoform | 0 | |
| 1,1,2,2-tetrachloroethane | 0 | Yes |
| 1,3-dichlorobenzene | 5.55E-13 | |
| 1,4 dichlorobebezne | 5.55E-13 | |
| benzyl chloride | 0 | |
| 1,2-dichlorobenzene | 5.55E-13 | |
| 1,2,4-trichlorobenzene | 0 | |
| hexachloro-1,3-butadiene | 0 | |
| carbon disulfide | 2.76E-12 | |
| Acrolein | 1.99E-11 | Yes |
| Acetone | 1.91E-13 | Yes |
| Isopropanol | 5.09E-12 | Yes |
| MTBE | 0 | Yes |
| vinyl acetate | 3.16E-11 | |
| MEK | 1.20E-12 | Yes |
| ethyl acetate | 1.60E-12 | Yes |
| Tetrahydrofuran | 1.61E-11 | |
| methyl methacrylate | 5.25E-11 | |

| | | |
|---|---|---|
| 1,4-dioxane | 3.83E-11 | |
| 4-methyl-2-pentanone | 1.27E-11 | Yes |
| 2-hexanone | 9.10E-12 | Yes |

**Table S2: VOC concentrations (ppbv) measured in the industrial area in Nanjing. VOC concentrations (ppbv) observed in**

**previous studies in Nanjing are also listed.**

[revised manuscript text omitted]

**S1. Source apportionment of VOCs**

970   Figure S1 shows the source profile of summertime VOCs. The resolved factors were identified as biomass/biofuel burning, LPG/NG usage, gasoline evaporation, gasoline vehicle exhaust, diesel vehicle exhaust, industrial production, paint solvent usage, and biogenic source. Factor 1 was characterized by high concentrations of ethane and ethylene. These compounds are tracers of incomplete combustion emitted from vehicle exhaust and biomass/biofuel burning (An et al.,

975   2017). Benzene, toluene, pentane, and decane concentrations were low in factor 1, therefore, it was identified as biomass/biofuel burning. Factor 2 was distinguished by a significant presence of LPG/NG VOCs propane, isobutene, and n-butane (Shao et al., 2016). So, factor 2 was identified as LPG/NG usage. Factor 3 was dominated by high concentrations of isopentane, n-pentane, and MTBE. Therefore, factor 3 was identified as gasoline evaporation (Song et al.,

980   2018; Wang et al., 2016).  Factor 4 possessed high concentrations of vehicle exhaust VOCs benzene and toluene (Song et al., 2018). These VOCs are also emitted by industrial processes. But the contribution of benzene was several folds higher than toluene. Therefore, factor 4 was related to vehicle exhaust emission and it was assigned to gasoline vehicle exhaust (An et al., 2017). Factor 5 was characterized by high concentrations of acetylene, n-heptane, and decane.

985   These are related to vehicle emissions. As diesel engines produce more acetylene than gasoline engines (Song et al., 2018; An et al., 2017), factor 5 was attributed to diesel vehicle exhaust. Factor 6 was dominated by toluene and the sampling site was beside an industrial area. So, we identified this factor as industrial production. Due to the high contribution of o-xylene, m,p-xylene, ethylbenzene and styrene, factor 7 was assigned to paint solvent usage sources (Li et al.,

990   2018). Factor 8 was attributed to the biogenic source, which was mainly distinguished by a high concentration of isoprene (Song et al., 2018).

During autumn, the possible VOC sources were biomass/biofuel burning, multiple sources, gasoline vehicle exhaust, vehicle emission, LPG/NG usage, paint solvent usage and gasoline evaporation (Fig.S2). Factor 1 was represented by a high concentration of ethane and ethylene,

995   so, it was identified as a biomass/biofuel burning source (An et al., 2017). Factor 2 was dominated by isoprene, n-heptane, decane, and acetylene. Among these compounds, isoprene is mainly emitted by trees and the rest of the compounds are related to diesel vehicle exhaust emission. Therefore, Factor 2 was identified as multiple sources. Factor 3 was identified as

gasoline vehicle exhaust due to the high contribution of benzene and toluene. In factor 3, benzene was several folds higher than toluene. The 4[th] factor was mainly composed of vehicle emission-related compounds 2-methyl pentane, n-hexane, n-heptane, n-pentane, and isopentane, therefore, identified as vehicle emission (Song et al., 2018). Factor 5 was assigned to LPG/NG usage as propane, isobutene, and n-butane were the main contributors (Shao et al., 2016). Factor 6 was characterized by a high concentration of o-xylene, m,p-xylene, ethylbenzene and styrene, which are typical tracers of paint solvent usage sources. Factor 7 was identified as gasoline evaporation, it was dominated by high concentrations of isopentane, n-pentane, and MTBE (Song et al., 2018).

During winter, the source factors were gasoline vehicle exhaust, vehicle exhaust, gasoline evaporation, biomass/biofuel burning, multiple sources, LPG/NG usage, and paint solvent usage (Fig. S3). Factor 1 was assigned to gasoline vehicle exhaust. It was dominated by benzene and toluene; the contribution of benzene was twice of toluene. Factor 2 was characterized by isobutene, n-butane, acetylene, ethylene, ethane, n-heptane and decane. Isobutene and n-butane are related to LPG/NG usage. But, the contribution of propane was zero in factor 2. Acetylene, ethylene, and ethane are emitted from combustion sources like vehicle exhaust and biomass burning. Decane and n-heptane are also related to vehicle emissions. By considering the above information, factor 2 was identified as vehicle exhaust. Factor 3 was characterized by high concentrations of isopentane and n-pentane, therefore, identified as a gasoline evaporation source. Factor 4 was characterized by a high contribution of ethylene and ethane; therefore, identified as a biomass/biofuel burning source. Factor 5 was characterized by high concentrations of isoprene, propane, n-hexane and n-heptane. Propane is related to LPG/NG usage, isoprene is mainly emitted from trees (evergreen trees in winter), and n-hexane and n-heptane are related to vehicle emission. By considering the above information, factor 5 was assigned to multiple sources. Factor 6 was dominated by high concentrations of propane. Therefore, it was identified as LPG/NG usage. Factor 7 was identified as paint solvent usage due to the high contribution of o-xylene, m,p-xylene, ethylbenzene and styrene (Zhang et al., 2018; Song et al., 2020).

During spring, the possible VOC sources were biomass/biofuel burning, paint solvent usage, multiple sources, gasoline evaporation, gasoline vehicle exhaust, LPG/NG usage, and diesel

vehicle exhaust (Fig. S4). Factors 1 was identified as a biomass/biofuel burning source for the

1030 high loading of ethylene and ethane and relatively lower contribution from the vehicle emission-related compounds. Due to the high contribution of o-xylene, styrene, m,p-xylene, and ethylbenzene, factor 2 was assigned to paint solvent usage sources (Li et al., 2018). Factor 3 had a high contribution of isoprene, n-hexane, n-heptane, decane, MTBE, toluene, ethylbenzene, and o-xylene. Therefore, factor 3 was identified as multiple sources. Factor 4 was represented by a

1035 high concentration of isopentane, n-pentane, and MTBE. Therefore, factor 4 was identified as gasoline evaporation. Factor 5 was represented by high concentrations of benzene, therefore, identified as gasoline vehicle exhaust. Factor 6 was assigned to LPG/NG usage due to the high contribution of propane, n-butane, and isobutane (Shao et al., 2016). Factor 7 was identified as diesel vehicle exhaust due to the high contribution of acetylene, n-heptane, and decane.

[Figure]

1040

**Figure S1: Source profile of VOCs during summer in Nanjing industrial area. Bars and dots represent the concentrations and percentages of the compounds, respectively.**

[Figure]

1043

**Figure S2: Source profile of VOCs during autumn in Nanjing industrial area. Bars and dots represent the concentrations and percentages of the compounds, respectively.**

[Figure]

1046

**Figure S3: Source profile of VOCs during winter in Nanjing industrial area. Bars and dots represent the concentrations and percentages of the compounds, respectively.**

1047

1048

[Figure]

1049

**Figure S4: Source profile of VOCs during spring in Nanjing industrial area. Bars and dots represent the concentrations and**

1051 **percentages of the compounds, respectively.**

[Figure]

**Figure S5: O₃ isopleth diagram on a high O₃ episode day (July 29 2018) in Nanjing industrial area.**

1055

1060

1065

---

## Author Response (AR2)

Dear Editor,

Many thanks for sending us further comments from the reviewers. The point-by-point replies to the comments are presented below. The revised manuscript with track changes is also added at the end of the document.

Best regards,
Authors

**Further comments by reviewers**

1. **The authors had been asked to comment on whether the measurement period from April-May 2020 was representative of a typical spring in Nanjing due changes in operations during the Covid pandemic. The authors response was that measurements conducted in the spring of 2020 were not impacted by Covid, and instead that they were unable to conduct measurements in the spring of 2019 due to the pandemic. The Covid pandemic first occurred in December of 2019, could the authors clarify this timeline along with any potential impacts on local emissions during the measurement time periods?**

Authors' response: We apologize for misinterpreting the reviewer's comment and submitting an irrelevant response to it. We also apologize for the earlier response's incorrect timeline. Nanjing city implemented a COVID-19 lockdown between February 03 and 19, 2020. From February 20, 2020, all economic activities, such as public transport and industry, were fully operational. As a result, the measurement period from April to May 2020 was reflective of a typical spring in Nanjing, and the pandemic had no impact on local emissions during the measurement periods.

2. **The manuscript would benefit from additional editing for grammar.**

Authors' response: Grammatical errors have been corrected in the manuscript. The revised manuscript with track changes is added below.

[revised manuscript text omitted]